# Expression of different L1 isoforms of *Mastomys natalensis* papillomavirus as mechanism to circumvent adaptive immunity

Yingying Fu[1‡], Rui Cao[1], Miriam Schäfer[1], Sonja Stephan[1],
Ilona Braspenning-Wesch[1], Laura Schmitt[1§], Ralf Bischoff[2], Martin Müller[3],
Kai Schäfer[1#], Sabrina E Vinzón[1†¶], Frank Rösl[1†*], Daniel Hasche[1†*]

[1]Division of Viral Transformation Mechanisms, Research Program 'Infection, Inflammation and Cancer', German Cancer Research Center, Heidelberg, Germany; [2]Division of Functional Genome Analysis, Research Program 'Functional and Structural Genomics', German Cancer Research Center, Heidelberg, Germany; [3]Research Group Tumorvirus-specific Vaccination Strategies, Research Program 'Infection, Inflammation and Cancer', German Cancer Research Center, Heidelberg, Germany

*For correspondence:
f.roesl@dkfz.de (FR);
d.hasche@dkfz.de (DH)

[†]These authors contributed equally to this work

Present address: [‡]Division of Infectious Diseases, Anschutz Medical Campus, University of Colorado School of Medicine, Aurora, United States; [§]Catalent Pharma Solutions, Gammelsbacher, Eberbach, Germany; [#]AbbVie Deutschland GmbH & Co. KG, Global Preclinical Safety, Development Sciences, Knollstrasse, Germany; [¶]Laboratory of Molecular and Cellular Therapy, Fundación Instituto Leloir, IIBBA-CONICET, Buenos Aires, Argentina

Competing interests: The authors declare that no competing interests exist.

**Abstract** Although many high-risk mucosal and cutaneous human papillomaviruses (HPVs) theoretically have the potential to synthesize L1 isoforms differing in length, previous seroepidemiological studies only focused on the short L1 variants, co-assembling with L2 to infectious virions. Using the multimammate mouse *Mastomys coucha* as preclinical model, this is the first study demonstrating seroconversion against different L1 isoforms during the natural course of papillomavirus infection. Intriguingly, positivity with the cutaneous MnPV was accompanied by a strong seroresponse against a longer L1 isoform, but to our surprise, the raised antibodies were non-neutralizing. Only after a delay of around 4 months, protecting antibodies against the short L1 appeared, enabling the virus to successfully establish an infection. This argues for a novel humoral immune escape mechanism that may also have important implications on the interpretation of epidemiological data in terms of seropositivity and protection of PV infections in general.

## Introduction

Human Papillomaviruses (HPVs) are widely distributed in nature and more than 220 types were sequenced up to date (PaVE: Papillomavirus Episteme). They cannot only be divided in mucosal and cutaneous types (*Bzhalava et al., 2013*), but also whether the infection is acquired, for example via sexual intercourse (as for high-risk genital HPVs) (*Gravitt and Winer, 2017*) or whether a commensal cohabitation (as for cutaneous HPVs) occurred shortly after birth (*Antonsson et al., 2003*; *Weissenborn et al., 2009*). Depending on environmental factors (e.g. chronic UV exposure) (*Rollison et al., 2019*; *Uberoi et al., 2016*), the individual immune status (e.g. systemic immunosuppression) (*Reusser et al., 2015*; *Vinzón and Rösl, 2015*) or genetic predispositions (e.g. EVER1/2 mutations in *Epidermodysplasia verruciformis* patients) (*de Jong et al., 2018*), commensal cutaneous papillomaviruses can induce hyperproliferative lesions (e.g. actinic keratosis) which may progress to squamous cell carcinomas (SCCs) (*Hasche et al., 2018*).

The African multimammate rodent *Mastomys coucha* represents a unique model system to investigate the consequences of a natural PV infection in the context of skin carcinogenesis (*Hasche and*

**eLife digest** Cancer is not one disease but rather a collection of disorders. As such there are many reasons why someone may develop cancer during their lifetime, including the individual's family history, lifestyle and habits. Infections with certain viruses can also lead to cancer and human papillomaviruses are viruses that establish long-term infections that may result in cancers including cervical and anal cancer, and the most common form of cancer worldwide, non-melanoma skin cancer.

The human papillomavirus, or HPV for short, is made up of DNA surrounded by a protective shell, which contains many repeats of a protein called L1. These L1 proteins stick to the surfaces of human cells, allowing the virus to get access inside, where it can replicate before spreading to new cells. The immune system responds strongly to HPV infections by releasing antibodies that latch onto L1 proteins. It was therefore not clear how HPV could establish the long-term infections and cause cancer when it was seeming being recognized by the immune system.

Now, Fu et al. have used the Southern multimammate mouse, *Mastomys coucha*, as a model system for an HPV infection to uncover how papillomaviruses can avoid the immune response. This African rodent is naturally infected with a skin papillomavirus called MnPV which, like its counterpart in humans, can trigger the formation of skin warts and malignant skin tumors.

Fu et al. took blood samples from animals that had been infected with the virus over a period of 76 weeks to monitor their immune response overtime. This revealed that, in the early stages of infection, the virus made longer-than-normal versions of the L1 protein. Further analysis showed that these proteins could not form the virus's protective shell but could trigger the animals to produce antibodies against them. Fu et al. went on to show that the antibodies that recognized the longer variants of L1 protein where "non-neutralizing", meaning that could not block the spread of the virus, which is a prerequisite for immunity. It was only after a delay of four months that the animals started making neutralizing antibodies that were directed against the shorter L1 proteins that actually makes up the virus's protective coat.

These findings suggest that virus initially uses the longer version of the L1 protein as a decoy to circumvent the attention of the immune system and provide itself with enough time to establish an infection. The findings also have implications for other studies that have sought to assess the success of an immune response during a papillomavirus infection. Specifically, the delayed production of the neutralizing antibodies means that their presence does not necessarily indicate that a patient is not already infected by a papillomavirus that in the future may cause cancer.

---

*Rösl, 2019*). The animals become infected with *Mastomys natalensis* papillomavirus (MnPV) soon after birth (*Schäfer et al., 2011*) and seroconversion against viral proteins can be detected shortly afterwards (*Schäfer et al., 2010*). MnPV is a typical cutaneous PV that resembles human β-types by lacking an E5 open-reading frame (ORF) (*Tan et al., 1994*). Characterization of the viral transcriptome in productive lesions revealed a complex splicing pattern with different promoters and transcriptional start sites (*Salvermoser et al., 2016*), also described for some HPV types (*Sankovski et al., 2014*; *Wang et al., 2011*) or for the mouse papillomavirus type 1 (MmuPV1) (*Xue et al., 2017*). Most of these transcripts are polycistronic, allowing (at least hypothetically) the translation of several different ORFs (*Salvermoser et al., 2016*).

Using *Mastomys coucha* as a preclinical model, we could show that immunization with MnPV virus-like-particles (VLPs) induces a long-lasting neutralizing antibody response that completely prevents the appearance of skin lesions both under immunocompetent and immunosuppressed conditions (*Vinzón et al., 2014*). Furthermore, *Mastomys coucha* also represents a paradigm for SCC development in the context of MnPV infection and UV exposure, thereby reflecting many aspects found in humans where a 'hit-and-run' mechanism during carcinogenesis is supposed (*Hasche et al., 2017*; *Hasche et al., 2018*).

Virions of PVs consist of 72 pentamers of the major (L1) protein together with up to 72 molecules of the minor (L2) capsid protein (*Buck et al., 2013*; *Hagensee et al., 1993*; *Wang and Roden, 2013*). The L1 protein has the capability to spontaneously form regular structures (capsomers), triggered by a thermodynamically favored self-assembly process (*McManus et al., 2016*). Due to their

repetitive structures, PV particles are very immunogenic and induce the generation of neutralizing antibodies that block viral entry into the host cell via binding to conformational epitopes on the capsid (*Kwak et al., 2011*; *Wang and Roden, 2013*).

Considering the cross-talk between viral infections and the immune system, PVs have developed multiple strategies to escape from immune surveillance (*Bordignon et al., 2017*). While there is plenty of information about how innate immunity as the first line of defense is circumvented (*Christensen, 2016*; *Smola et al., 2017*), less is known about the humoral immune response in terms of generation of protecting antibodies during the natural course of a PV infection.

In the present study, we show that MnPV, as a rodent equivalent for cutaneous PVs in humans, induces a strong seroconversion in its natural host early after infection. However, the raised antibodies are non-neutralizing and directed against a longer isoform of the L1 protein which is unable to assemble into viral particles. Only after a delay of around 4 months after infection, protecting antibodies appear. This argues for a novel PV immune escape mechanism, probably providing a selective advantage to establish an efficient infection. We characterized this mechanism in greater detail since it may also have important implications in understanding the humoral immune response during a normal infection cycle in general.

## Results

### Alternative translation initiation codons of the PV L1 ORF

Based on two previous studies comparing the presence of initiation codons within the papillomavirus L1 ORF (*Joh et al., 2014*; *Webb et al., 2005*), their position was aligned according to the PV genera derivation (*Bzhalava et al., 2015*; *Van Doorslaer et al., 2013*; *Figure 1*). Notably, alternative ATGs can be found in various mucosal 'high-risk' HPV types such as 16, 18, 45, 52, 56, 58, but not in 'low-risk' types such as HPV6, 11, 40, 42, 43, 44, respectively (*Webb et al., 2005*). Additional in-frame initiation codons can also be detected in cutaneous HPV types of several genera such as HPV1, 2, 8, 38, 41, 57 and 77, respectively, of which HPV8 and HPV38 are considered to be 'high-risk' cutaneous HPVs (*Rollison et al., 2019*; *Tommasino, 2017*). Accordingly, due to the presence of potential alternative translation initiation sites, different L1 isoforms could be translated. As shown in *Figure 1*, almost all outlined PV L1 proteins harbor a consensus $Wx_7YLPP$ motif within the N-terminal region (*Joh et al., 2014*), independently from PV genus or cancer risk assessment, while the remaining N-terminal sequences are not very conserved. For the majority of PV types, the nearest methionine codon to this motif is located one to three amino acids upstream of the tryptophan (W) in the consensus motif. However, there are also exceptions from this rule since HPV31, 35 and 51, for instance, harbor one additional ATG followed by an interspersed in-frame TAA stop codon, thereby probably preventing the synthesis of additional L1 isoforms (*Webb et al., 2005*). Intriguingly, the MnPV L1 ORF also contains three alternative ATGs, which are located at nucleotide positions (nt) 5704, 5725 and 5797, potentially leading to the expression of $L1_{LONG}$, $L1_{MIDDLE}$ and $L1_{SHORT}$ proteins, respectively.

### Anti-$L1_{LONG}$ and anti-$L1_{SHORT}$ seroresponses emerge at different time points after infection

To examine serological responses against the three putative MnPV L1 isoforms, 60 naturally infected animals were monitored during different stages of viral infection (682 sera in total) encompassing an age between 8 and 76 weeks. Since most serological detection methods developed to date are based on the $L1_{SHORT}$ isoform, firstly we examined seroconversion against $L1_{SHORT}$ by glutathione S-transferase (GST)-capture ELISA (*Kricker et al., 2020*; *Sehr et al., 2002*; *Waterboer et al., 2009*). Notably, only few animals (8/60) exhibited measurable seroresponses against $L1_{SHORT,}$ initially at an age of 28 weeks. The mean seroreactivity per time point exceeded the cut-off earliest at an age of 68 weeks (*Figure 2A*). Conversely, in 27.5% of the animals, broad seroresponses against MnPV $L1_{LONG}$ were already detectable as early as 8 weeks of age which increased to 52.5% of the animals at 76 weeks (*Figure 2B*). In the main comparison at the latest time point where most animals were still alive (68 weeks), a significant difference was observed (p<0.001, two-tailed McNemar's test, see *Figure 2—figure supplement 1*). Seroreactivity against $L1_{MIDDLE}$ shows a similar time course as $L1_{LONG}$ which is consistent with the correlation between both (*Figure 2C* and *Figure 2—figure*

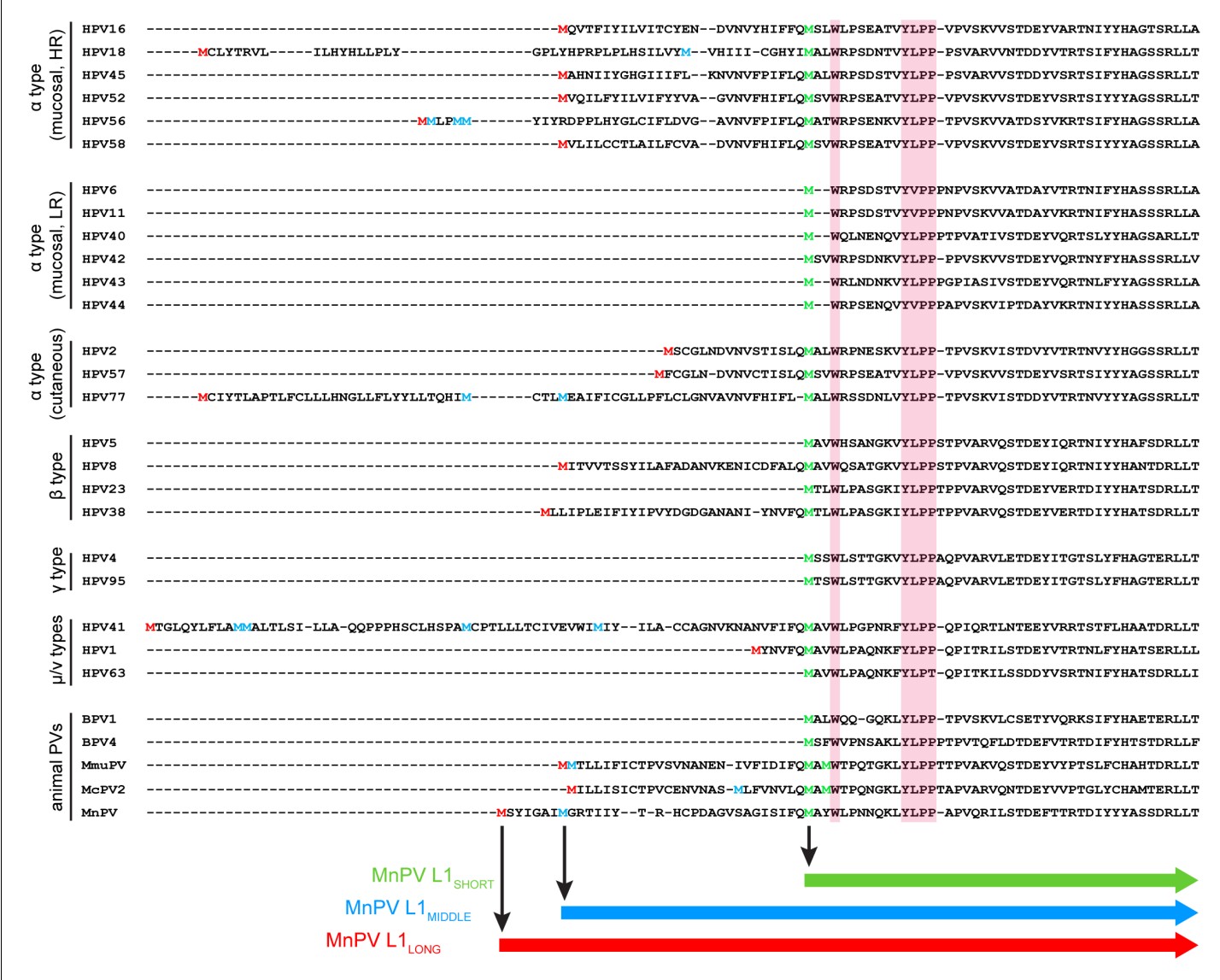

**Figure 1.** Alignment of L1 sequences from different PV types. N-terminal sequences of L1 proteins from 29 PV types were aligned using Clustal Omega. The highly conserved motif (Wx$_7$YLPP) is marked in pink boxes. The first methionine of L1 is marked in red. The last methionine upstream of the Wx$_7$YLPP motif is shown in green and methionines between the first and the last one are depicted in blue. In the case of MmuPV and McPV2, both methionines upstream of the conserved motif fit to the consensus sequence of L1$_{SHORT}$ and are therefore depicted in green.

supplement 2. Seroresponses against the E2 protein, which is involved in viral DNA replication (McBride, 2013) and considered as an early marker of infection (Schäfer et al., 2011), developed shortly after birth and increased during the study (Figure 2D). Conversely, seroconversion against the minor capsid protein L2 appeared only in a few of the animals and as late as seroconversion against L1$_{SHORT}$ (Figure 2E).

In order to exclude possible experimental bias merely working with a GST-fusion protein-based L1$_{SHORT}$ ELISA, additional ELISAs were performed using MnPV VLPs (derived from L1$_{SHORT}$) produced via baculovirus expression system (Christensen et al., 1994; Rose et al., 1993). In accordance with the GST-L1$_{SHORT}$ ELISA, serum responses against VLPs were absent in early infection stages and did not exceed the cut-off before an age of 20 weeks (Figure 2F). This suggests that anti-L1$_{LONG}$ antibodies during early infection fail to recognize epitopes of both GST-L1$_{SHORT}$ antigen and on the surface of intact VLPs. Accordingly, there is no correlation of anti-GST-L1$_{LONG}$ with anti-GST-L1$_{SHORT}$ or anti-VLP reactivity (Figure 3A and B). Conversely, a significant correlation between

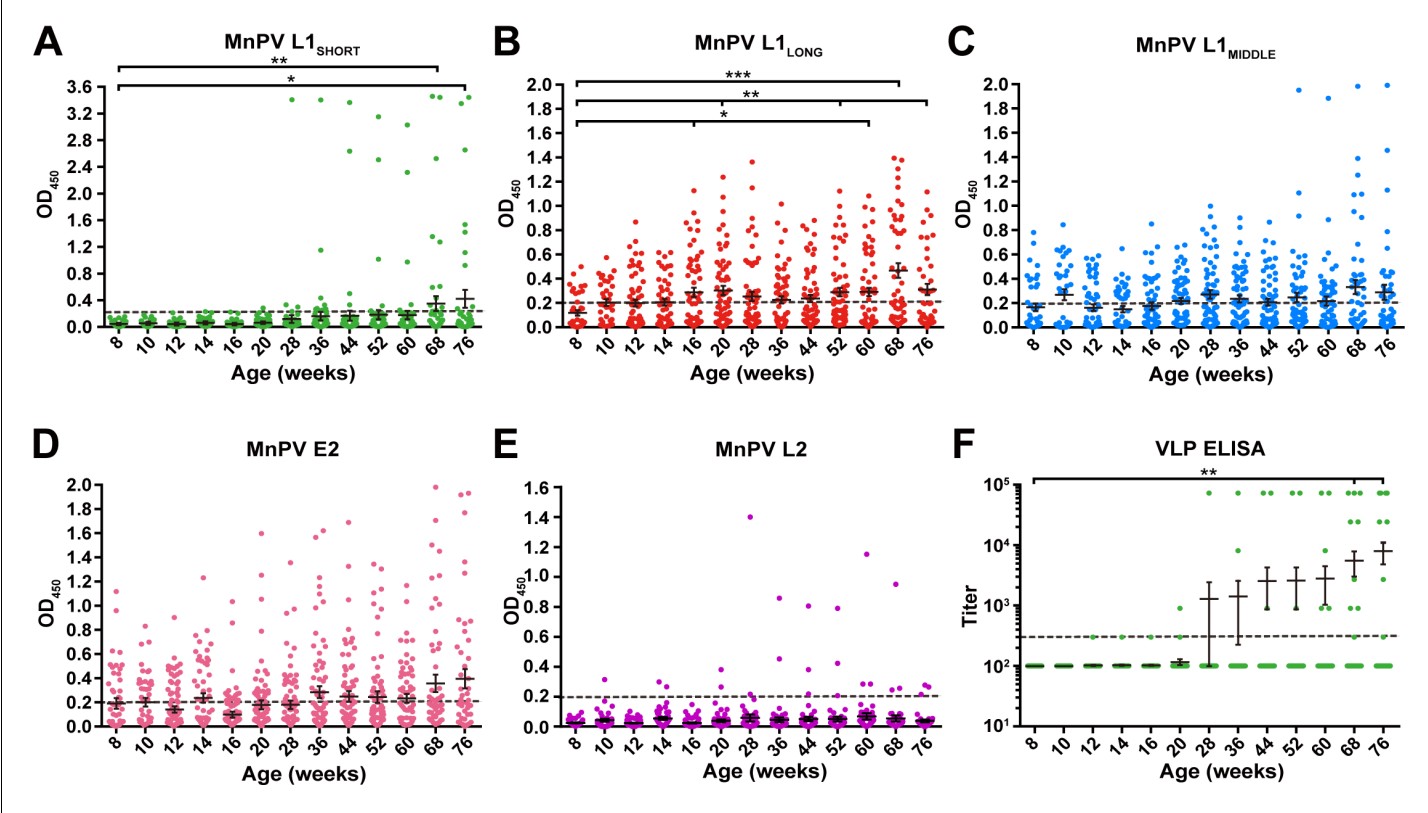

**Figure 2.** Seroreactivity against viral proteins in naturally MnPV-infected animals. Seroresponses of 682 sera from 60 animals measured by GST-ELISA against (A) $L1_{SHORT}$, (B) $L1_{LONG}$, (C) $L1_{MIDDLE}$, (D) E2 and (E) L2 GST-fusion proteins and F) VLP-ELISA. Dashed lines represent the methods' cut-off ($OD_{450}$ = 0.2 for GST-ELISA or titer of 300 for VLP-ELISA) (Mean ± SEM; 1-Way-ANOVA test, *p<0.05, **p<0.01, ***p<0.001).

The online version of this article includes the following figure supplement(s) for figure 2:

**Figure supplement 1.** Rate of $L1_{LONG}$ and $L1_{SHORT}$ positive animals.

**Figure supplement 2.** Additional correlation of seroreactivities measured by GST-ELISA.

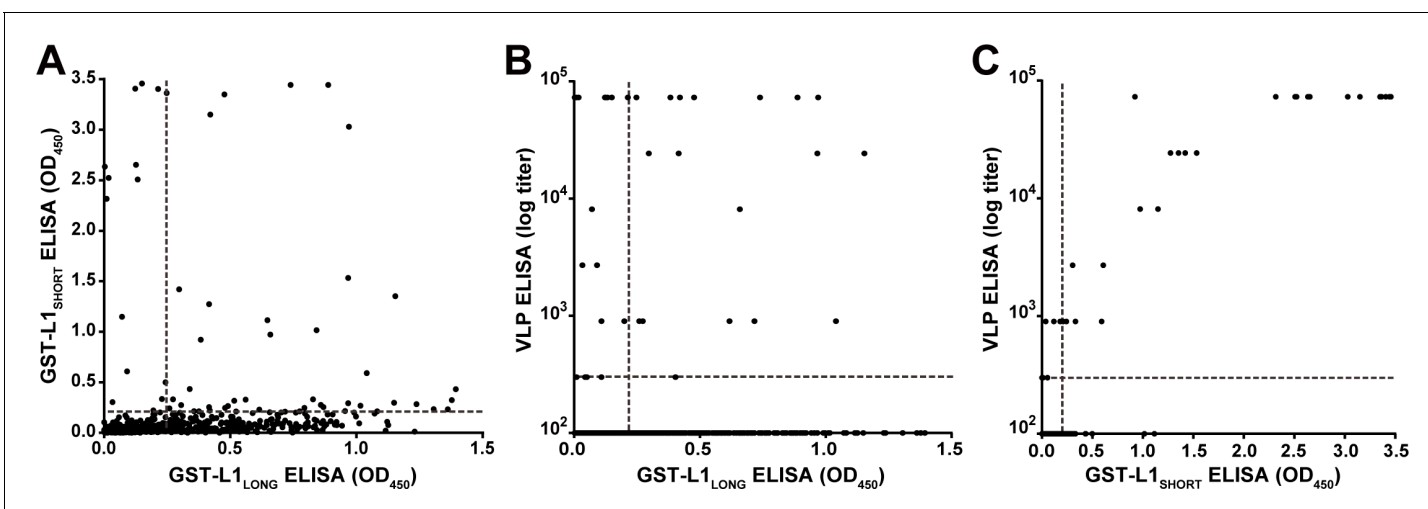

**Figure 3.** Correlation of GST- and VLP-ELISAs. (A) Correlation of seroreactivities against GST-$L1_{LONG}$ with seroreactivities against GST-$L1_{SHORT}$ (correlation coefficient, $R^2$ = 0.0261). (B) Correlation of GST-$L1_{LONG}$ ELISA with VLP-ELISA ($R^2$ = 0.0062). (C) Correlation of GST-$L1_{SHORT}$ ELISA with VLP-ELISA ($R^2$ = 0.8394). All graphs include all 682 sera taken during the study. Dashed lines indicate the methods' cut-offs.

GST-L1$_{SHORT}$ and VLP-ELISA (*Figure 3C*) strengthens the notion that the absence of a correlation between GST-L1$_{LONG}$ and L1$_{SHORT}$ ELISAs was indeed due to altered serological properties of L1 isoforms rather than due to different ELISA methodologies.

## Anti-L1$_{LONG}$ antibodies lack neutralizing capacity

Due to the different temporal order of seroconversion against L1$_{LONG}$, L1$_{SHORT}$ and VLPs, we reasoned that MnPV escapes from adaptive immunity to establish an efficient infection and to maintain a persistent life cycle, which is indicated by the increased seroresponse against MnPV E2 (*Figure 2D*). To get insight into this question, pseudovirion-based neutralization assays (PBNA) (*Pastrana et al., 2004*; *Roden et al., 1996*; *Vinzón et al., 2014*) were performed to monitor for the presence of protecting antibodies. As shown in *Figure 4A*, PBNA revealed a similar kinetics as previously demonstrated for the L1$_{SHORT}$ isoform, indicating that neutralizing antibodies in fact appeared delayed. This was further substantiated by correlating the serum titers measured by VLP-ELISA with data obtained by PBNA (*Figure 4B*). Conversely, all sera directed against L1$_{LONG}$ (but negative for L1$_{SHORT}$) lack neutralizing capability (*Figure 4C*).

## Mapping of immunodominant epitopes in MnPV L1

In order to identify epitopes within L1 recognized by the sera, synthetic linear 15-mer peptides with 14 residue overlaps were spotted on microarrays. Incubation with a *Mastomys* serum mix obtained from five tumor-bearing animals, possessing high titers against L1$_{LONG}$ and L1$_{SHORT}$ identified three immunogenic epitopes within the region homologous between L1$_{LONG}$ and L1$_{SHORT}$ (ITGHPLY, DYLGMSK and KRSLPASRN) (*Figure 5A and B*). Notably, two of them (ITGHPLY, DYLGMSK) coincide with the DE and FG loops (*Figure 5C*; *Bissett et al., 2016*) that form conformational epitopes on the surface of HPV virions (*Li et al., 2017*; *Zhang et al., 2016*) and are known to be highly immunogenic.

## Epitope characterization of anti-L1$_{LONG}$ and anti-L1$_{SHORT}$ antibodies

To further elucidate why anti-L1$_{LONG}$ antibodies lack neutralizing capacity, we dissected the seroresponse against L1 with respect to the 31 amino acids present only at the N-terminus of L1$_{LONG}$ (*Figure 1*). Analyzing 297 sera of 39 L1$_{LONG}$-positive animals by GST-ELISA, no detectable or only weak positivity could be measured against this L1$_{LONG}$aa1-31 (*Figure 6A*). In contrast, when extending

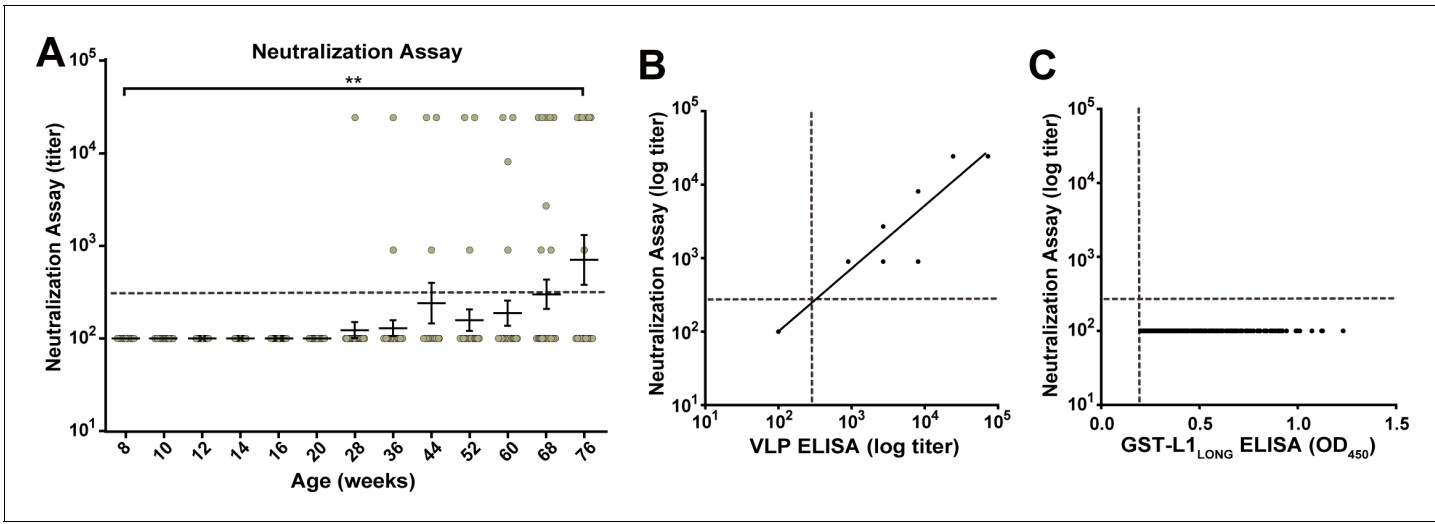

**Figure 4.** Neutralizing capacity of anti-L1$_{LONG}$ and anti-L1$_{SHORT}$ antibodies. (**A**) Neutralization assay for all L1$_{LONG}$-/L1$_{SHORT}$-positive sera (n = 294) from 60 naturally infected animals. (**B**) Correlation of VLP-ELISA titers and neutralizing titers of all L1$_{LONG}$-/L1$_{SHORT}$-positive sera (correlation coefficient, $R^2 = 0.9883$). The regression line represents a linear regression fit (Please note, that for both assays all sera were diluted in three-fold dilution steps. Since the titers are calculated from the dilution, data points of 294 different sera overlay when having the same titer in both assays). (**C**) Correlation of GST-L1$_{LONG}$ ELISA and neutralization assay for 234 L1$_{LONG}$-positive/L1$_{SHORT}$-negative sera ($R^2 = 0.0000$). Correlation analyses contain sera from animals representing the complete age range. Dashed lines indicate the methods' cut-offs (OD$_{450}$ = 0.2 for GST-ELISA or titer of 300 for neutralization assay).

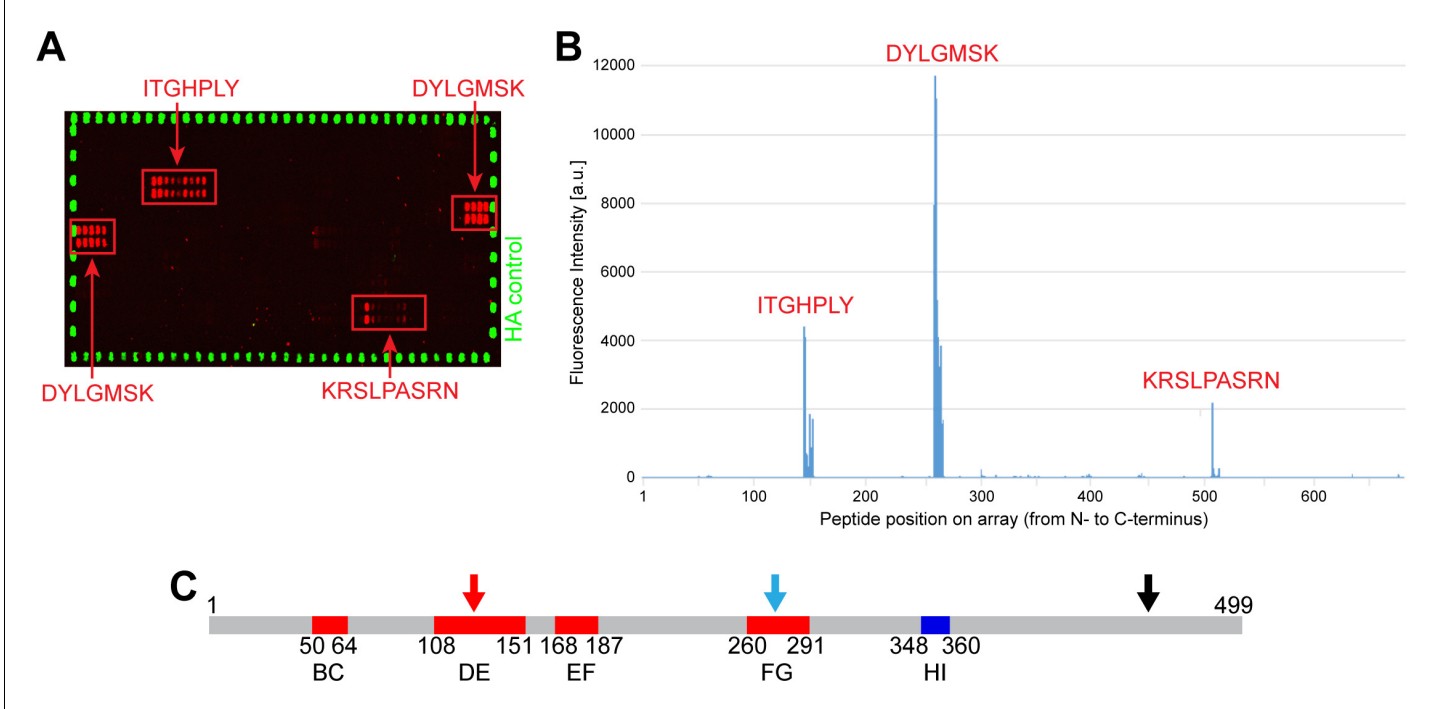

**Figure 5.** Peptide arrays identify known immunogenic epitopes in L1. (**A**) Synthetic 15-mer peptides with residue overlaps of 14 residues were spotted on microarrays and incubated with serum mix from five tumor-bearing animals with high titers against both L1 isoforms. Bound serum antibodies were detected with fluorophore-conjugated secondary antibodies. Positive regions (ITGHPLY, DYLGMSK and KRSLPASRN) are indicated and were mapped to their position in L1$_{LONG}$ (**B**). (**C**) Two of these regions (ITGHPLY and DYLGMSK) coincide with the DE and the FG loop, respectively (scheme shows MnPV L1$_{SHORT}$; see **Supplementary file 1**).

The online version of this article includes the following figure supplement(s) for figure 5:

**Figure supplement 1.** Immunogenicity prediction of L1$_{LONG}$.

these 31 amino acids of L1$_{LONG}$ to 41 residues (which includes nine residues of L1$_{SHORT}$), the number of seropositive animals strikingly increased (**Figure 6B**), whereas such a reactivity could not be observed in sera from MnPV-free animals vaccinated with VLPs (made from L1$_{SHORT}$) obtained in a previous study (**Vinzón et al., 2014**; **Figure 6C**). Correlating reactivities between L1$_{LONG}$ and L1$_{LONG}$aa1-41 (**Figure 6D**) shows that actually all sera which are positive for L1$_{LONG}$aa1-41 are also positive for L1$_{LONG}$, (which is not the case for L1$_{SHORT}$, see **Figure 6—figure supplement 1**). It is therefore reasonable to assume that this antibody population is likely arising from exposure of the immune cells to the L1$_{LONG}$ antigen.

These data, together with the finding that the first linear epitope recognized on the peptide array is located in L1$_{SHORT}$ (**Figure 5**), indicate that a conformational epitope is spanning L1$_{LONG}$ and L1$_{SHORT}$, and represents both the most immunogenic epitope in L1$_{LONG}$ and a major immunogen in early stages of infection. Moreover, an algorithm that predicts antigenic sites on proteins (**Kolaskar and Tongaonkar, 1990**), calculates the abovementioned DE loop (determinant no. 8) and partially the FG loop (determinant no. 12) (**Figure 5—figure supplement 1**) detected by the serum mix (see **Figure 5**) and also predicts an N-terminal epitope between residues 11 and 34 (determinant no. 1).

To further characterize the antigenic properties and to prove that a conformational epitope is formed at the N-terminus of L1$_{LONG}$, ELISAs with denatured antigens (VLPs or GST fusion proteins) were performed. For this purpose, a panel of monoclonal antibodies raised against MnPV VLPs was used. These antibodies differ in terms of neutralization and sensitivity in VLP and GST-ELISAs (**Supplementary file 2**). Of these, only mAb 2E2, 2D11 and 3H8 showed high reactivity against native VLPs (**Figure 7A**, grey, yellow, and green lines), indicative for the recognition of conformational epitopes. Conversely, mAb 2D6 and 5E5 that possess low binding to intact VLPs were expected to represent antibodies recognizing linear epitopes (**Figure 7A**, blue and purple lines). To

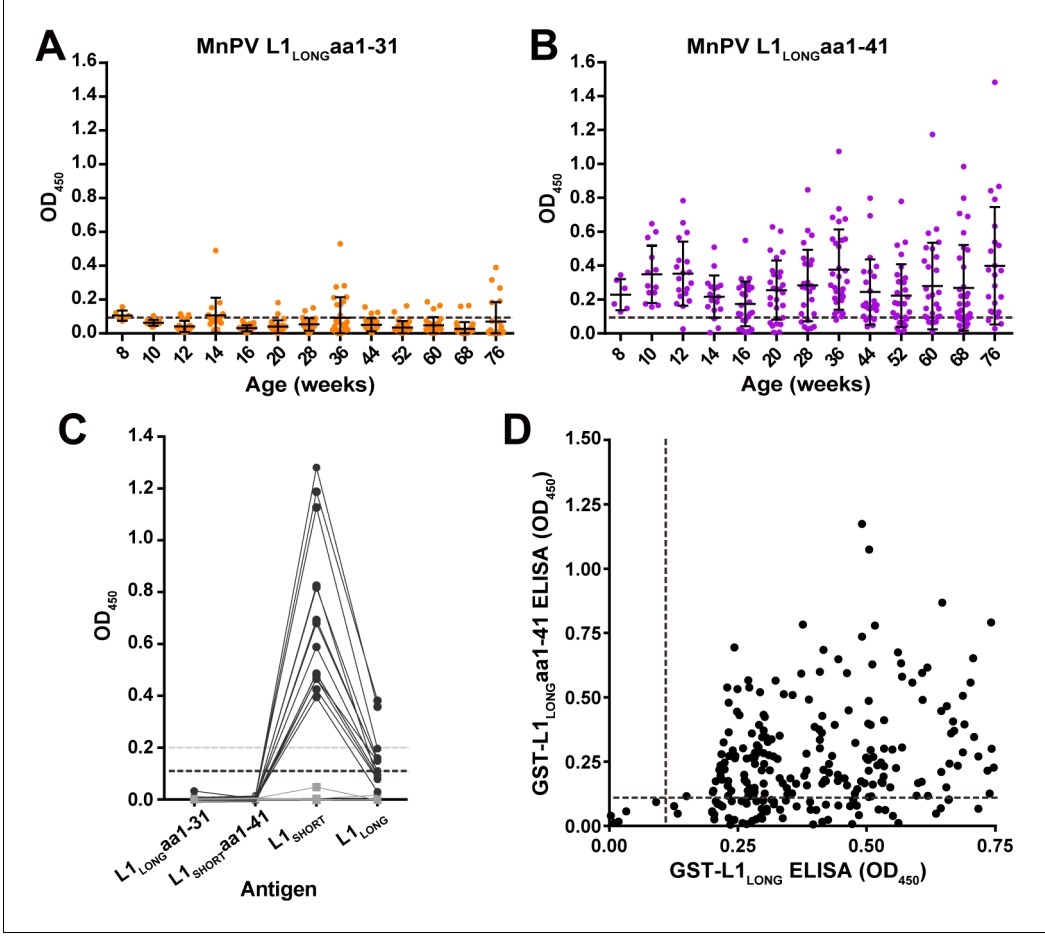

**Figure 6.** Seroreactivity against the N-terminus of L1$_{LONG}$ measured by GST-ELISA. (A) Seroreactivity against the 31 aa exclusive for L1$_{LONG}$ (60 animals, 682 sera) and (B) Against 41 aa of the N-terminus of L1$_{LONG}$ (39 L1$_{LONG}$-positive animals, 297 sera). Dashed lines represent the cut-off (0.11) based on virus-free animals. (C) Sera from MnPV-free animals (14 sera, black dots) vaccinated with VLPs (made from L1$_{SHORT}$) and six pre-immune sera (grey squares) from a previous study (*Vinzón et al., 2014*) measured in the different L1 GST-ELISAs. Dashed lines indicate the cut-offs (grey: OD$_{450}$ = 0.2 for L1$_{SHORT}$ and L1$_{LONG}$; black: OD$_{450}$ = 0.11 for L1$_{LONG}$aa1-31 and L1$_{LONG}$aa1-41). (D) Correlation of ELISAs for GST-L1$_{LONG}$ and GST-L1$_{LONG}$aa1-41 (correlation coefficient, $R^2$ = 0.0996).

The online version of this article includes the following figure supplement(s) for figure 6:

**Figure supplement 1.** Additional correlation of seroreactivities measured by GST-ELISA.

test this assumption, we anticipated a reversed pattern upon denaturation where the latter antibodies can bind, while binding of mAb 2E2, 2D11 and 3H8 should be abrogated. As shown in *Figure 7B*, this was indeed the case (see also *Supplementary file 2*).

Sera from the collective of 60 naturally infected animals previously tested positive for L1$_{LONG}$ and L1$_{SHORT}$ were also tested in denatured-VLP ELISA (306 sera) and denatured-GST-L1 ELISA (281 sera). Interestingly, denaturation of both VLPs and GST-L1 antigens abrogated the reactivity of all sera (*Figure 7C–E*), suggesting that anti-L1$_{LONG}$ and anti-L1$_{SHORT}$ antibodies were indeed directed against conformational epitopes.

## L1$_{SHORT}$ but not L1$_{LONG}$ and L1$_{MIDDLE}$ can form VLPs and infectious pseudovirions

To test the different isoforms for their capability to form virus-like structures, the ORFs of L1$_{SHORT}$, L1$_{MIDDLE}$ and L1$_{LONG}$ were expressed in Sf9 insect cells by the use of recombinant baculoviruses. Different preparations were analyzed by CsCl density gradient centrifugation. For quality control, the

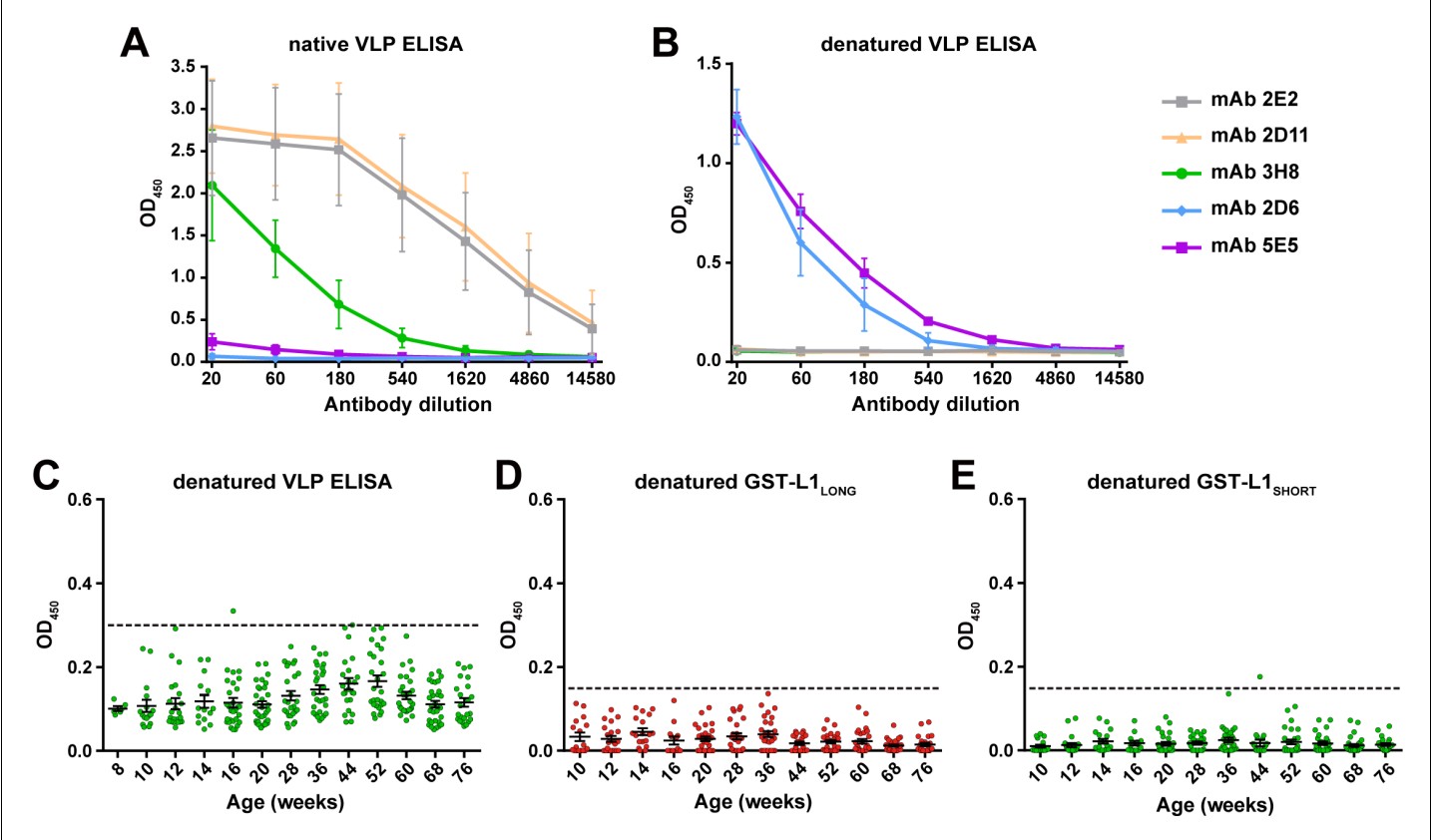

**Figure 7.** L1 denaturation abolishes recognition by naturally-raised antibodies. Binding efficacy of five monoclonal antibodies against (**A**) native VLPs and (**B**) VLPs denatured by heating (Mean ± SD, n = 3). (**C**) Seroresponse of *Mastomys* sera (306 sera positive for L1$_{LONG}$ and L1$_{SHORT}$) against denatured VLPs. (**D**) Seroresponse of *Mastomys* sera against denatured GST-L1$_{LONG}$. (**E**) Seroresponse of *Mastomys* sera against denatured GST-L1$_{SHORT}$. For (**E** and **D**), 281 sera from 40 animals positive for L1$_{LONG}$ and L1$_{SHORT}$ were measured. Dashed lines indicate the cut-offs (OD$_{450}$ = 0.3 for denatured VLP-ELISA; OD$_{450}$ = 0.15 for denatured GST-ELISA).

gradients' refractive indices were measured and corresponding fractions were analyzed via western blot, where all three L1 isoforms could be found, ranging between 55 and 70 kDa (*Figure 8—figure supplement 1*). To analyze the ability of the isoforms to form VLPs or similar structures, different fractions of the gradients were examined by EM. Considering L1$_{SHORT}$, highly concentrated and spherically shaped particles with sizes of 60 nm and clearly visible capsomers could be found (*Figure 8A*). Also in the lowest density fractions (e.g. fraction 11), particles with capsomer-like structures were detected. Conversely, inspecting the gradients of L1$_{LONG}$ and L1$_{MIDDLE}$, similar assemblies were absent, although many particles of different sizes (20–50 nm) were found (*Figure 8B and C*). However, it is difficult to classify these substructures as regular capsomers, indicating that L1$_{LONG}$ and L1$_{MIDDLE}$ were apparently not able to form correctly assembled VLPs under the same experimental conditions.

To examine whether the addition of L2, regularly present in mature infectious virions, can facilitate the formation of L1$_{LONG}$ or L1$_{MIDDLE}$ composed particles, pseudovirions were produced. 293TT cells were transfected with expression plasmids encoding L2 and the different L1 isoforms in conjunction with a reporter plasmid. The assembled structures were purified via Optiprep gradients and infection assays with different fractions were performed. In contrast to L1$_{SHORT}$ (*Figure 8D*), L1$_{LONG}$ and L1$_{MIDDLE}$ again did not yield virus-like structures even in the presence of L2 (*Figure 8E and F*). Moreover, while infectious L1$_{SHORT}$-based pseudovirions can be found in most of the gradient fractions, no signals of the reporter construct could be discerned for L1$_{LONG}$ and L1$_{MIDDLE}$. Luciferase signals could be measured in non-fractionated 293TT cell lysates, indicating that the transfection was successful for all three L1 isotypes.

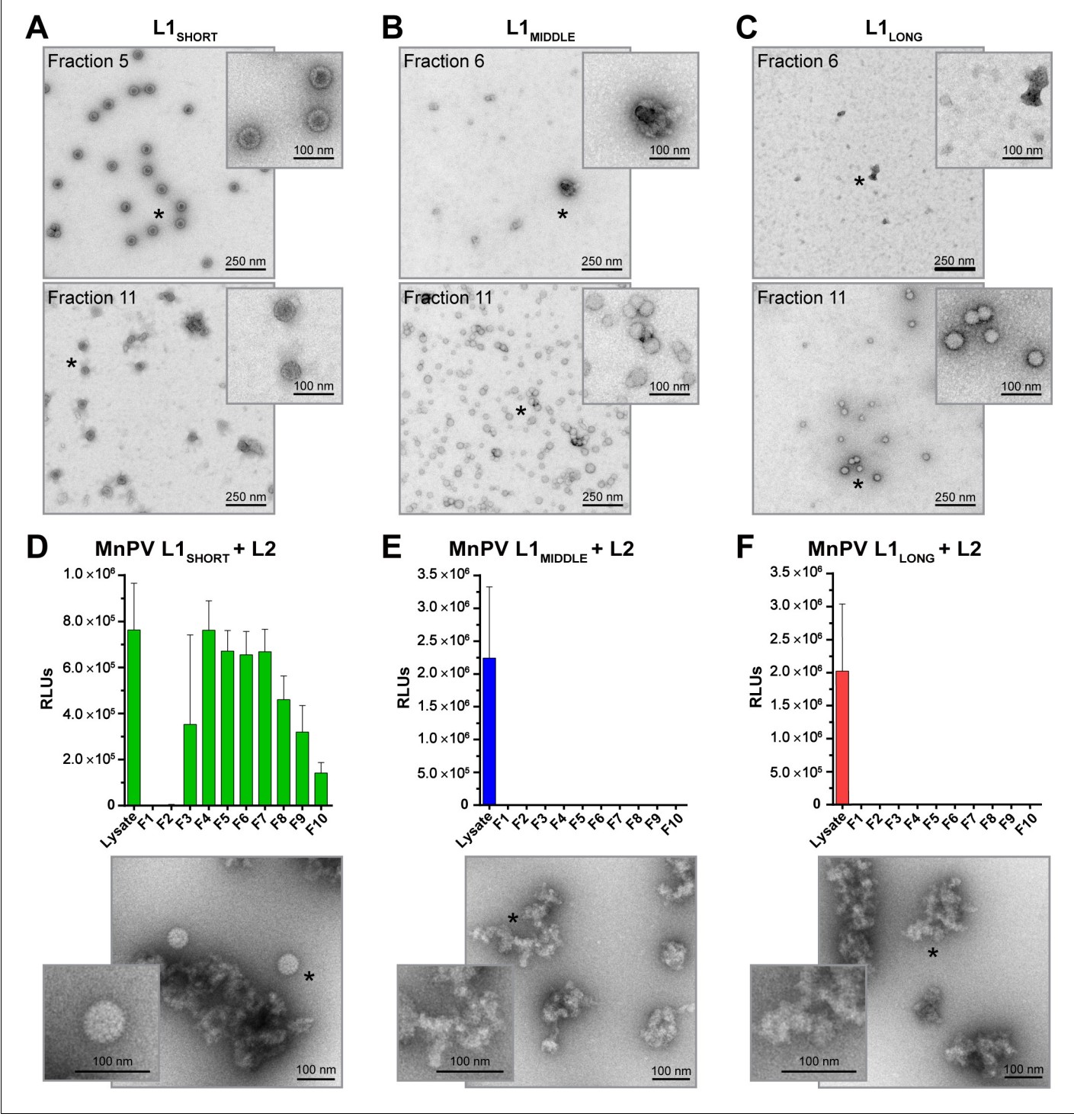

**Figure 8.** VLP and pseudovirion formation capacity of L1$_{SHORT}$, L1$_{MIDDLE}$ and L1$_{LONG}$. EM micrographs of peak fractions and the respective fractions of lowest densities of (A) L1$_{SHORT}$ (B) L1$_{MIDDLE}$ and (C) L1$_{LONG}$ at 16,000x magnification. Capacity to form infectious pseudovirions in presence of L2 was analyzed by infectivity assay and EM for (D) L1$_{SHORT}$ (E) L1$_{MIDDLE}$ and (F) L1$_{LONG}$. Note that high signals with unpurified lysate from PsV-producing cells result from co-expressed luciferase reporter protein (Mean ± SD, n = 3).

The online version of this article includes the following figure supplement(s) for figure 8:

**Figure supplement 1.** Production of L1 isoforms with the MultiBac baculovirus expression system.
**Figure supplement 2.** Pseudoatomic modeling of L1 proteins in the viral capsid.

## L1$_{SHORT}$ but not L1$_{LONG}$ can form high-MW structures in genuine host cells

To further investigate L1 isoform expression in their genuine host in vitro, *Mastomys*-derived fibroblasts (*Hasche et al., 2016*) were transfected either with plasmids exclusively encoding HA-tagged L1$_{SHORT}$, L1$_{MIDDLE}$ or L1$_{LONG}$ or with the polycistronic plasmid vL1 encoding the genuine viral L1 ORFs as found in natural transcripts. While serum mix from tumor-bearing animals was able to detect all L1 isoforms in immunofluorescence stainings, mAb 2D11 (which exclusively binds a conformational L1$_{SHORT}$ epitope, see *Figure 7D and E*, respectively) could only detect L1$_{SHORT}$ but not L1$_{MIDDLE}$ and L1$_{LONG}$, despite similar expression levels of all isoforms (see HA-tag) (*Figure 9A*). This indicates that in *Mastomys* cells only the L1$_{SHORT}$ can form structures involved in the formation of virus particles.

Western blot analysis of the corresponding cells showed similar expression of all HA-tagged L1 isoforms, which again could be detected by the serum mix (*Figure 9B*). Notably, only in L1$_{SHORT}$-transfected cells both mAb 2D11 and serum mix detected bands between 100 and 130 kDa and around 250 kDa, which disappeared in lysates treated with additional DTT, β-mercaptoethanol and extended heating prior to SDS-PAGE separation (*Figure 9C*). Their sizes correlate to L1-dimers and trimers and since mAb 2D11 does not detect linear epitopes, this suggests that only the L1$_{SHORT}$ isoform is able to form capsomer-like structures, which are at least partially structured in non-reducing conditions due to stabilizing properties of inter-capsomeric disulfide bridges (*Buck et al., 2005b*; *Sapp et al., 1998*).

Previous viral transcriptome analysis revealed the presence of three polycistronic transcripts (referred to as Q, R and S) that have the potential to encode both L1$_{LONG}$ and L1$_{SHORT}$ (*Salvermoser et al., 2016*). Using the most abundant of these (transcript Q) for prediction of the start codons' likelihood of being used for translation initiation (*Nishikawa et al., 2000*), it turned out that the ORFs of E1E4 (reliability index, RI = 0.43), L2 (RI = 0.42) and L1$_{LONG}$ (RI = 0.38) could be favored over L1$_{SHORT}$ (RI = 0.17) (*Supplementary file 3*). Indeed, consistent with the immunofluorescence, when transfecting cells with the polycistronic construct vL1, L1$_{LONG}$ as well as L1$_{SHORT}$ and its multimer band are detected, which confirms that both ORFs are functional and that L1$_{LONG}$, although unable to form capsomers, can be expressed from such a polycistronic construct (*Figure 9*).

## L1$_{LONG}$ and L1$_{SHORT}$ appear at different locations in MnPV-induced tumors

Considering the PV life cycle, virions are released by shedding of terminally differentiated cells at the uppermost layer (*stratum corneum*) of the epidermis. Here, L1 and L2 are assembled to capsomers and virus particles (*Figure 10A*). When staining MnPV-induced papillomas with serum that detects mature MnPV virions, virus particles can only be found in cornified structures above or within the epidermis and in islands of terminally differentiated keratinocytes (*Figure 10B*). This is consistent with the expression of L2, which also appears the earliest in nearly shed cells (*Figure 10C*). Conversely, using serum from mice immunized with the N-terminal peptide exclusive for L1$_{LONG}$, keratinocytes in the basal layer and the complete epithelium are positively stained (*Figure 10B*). This suggests that its expression takes place already during early infection phases long before viral particles are formed.

## Discussion

The non-random distribution of several ATG initiation codons within the L1 open reading frame (ORF) of certain human and animal papillomaviruses (*Figure 1*) potentially allows the translation of different L1 isoforms. Despite plenty of seroepidemiological studies on HPV, to our knowledge, seroresponses against different L1 isoforms have never been performed. Previous studies only focused on the shortest variant, known to efficiently form virus-like particles (VLPs) or, together with L2, infectious virions (*Buck et al., 2005b*).

The MnPV-infected rodent *Mastomys coucha* is a reliable preclinical model that mimics many aspects of the situation found in humans infected with cutaneous HPVs. Since the animals are immunocompetent and become naturally infected early in life (*Hasche and Rösl, 2019*; *Hasche et al.,*

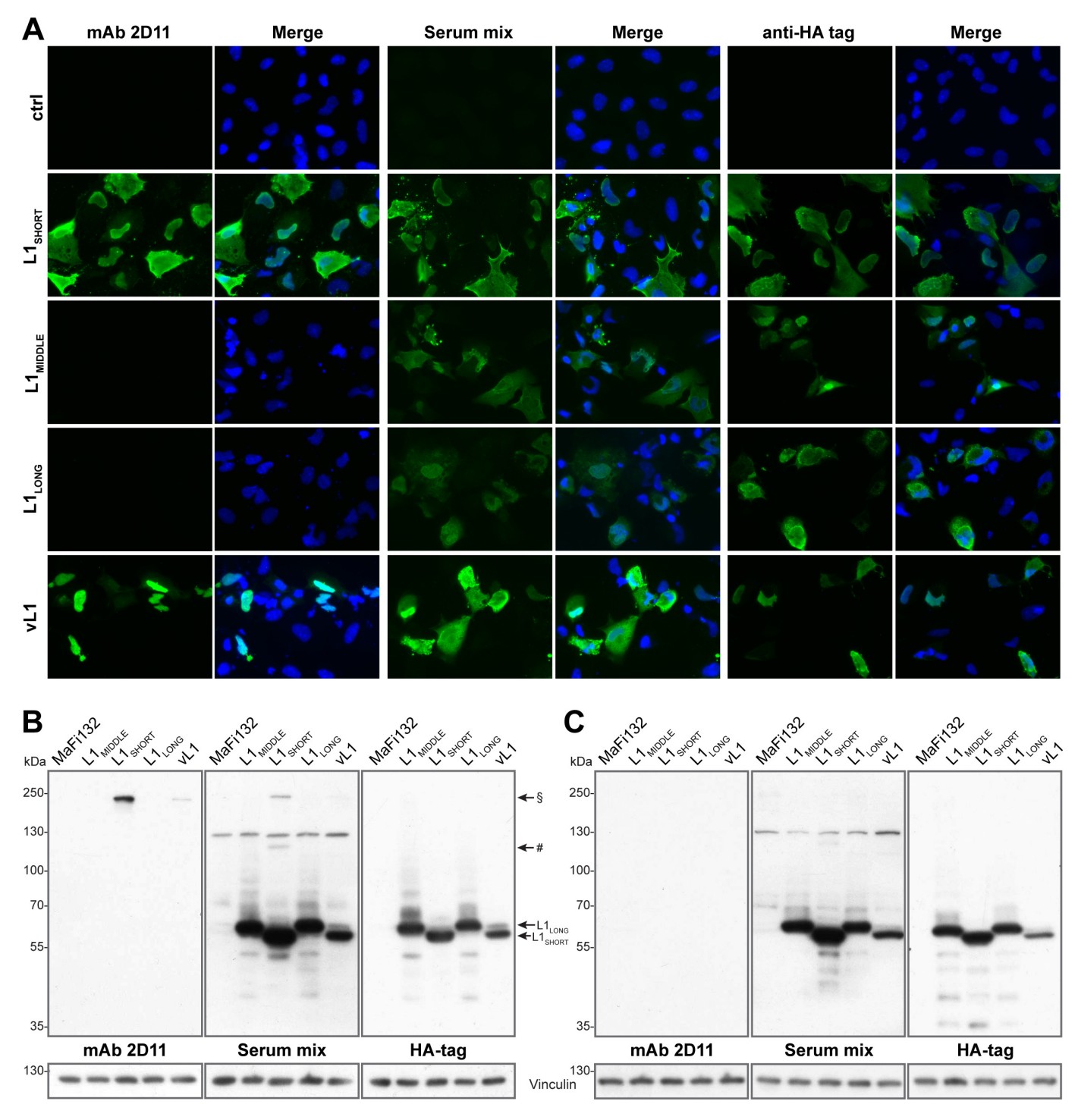

**Figure 9.** L1$_{SHORT}$ and L1$_{LONG}$ protein expose different epitopes when expressed in *Mastomys* cells. *Mastomys*-derived fibroblasts were transfected with HA-tagged L1$_{SHORT}$, L1$_{MIDDLE}$ or L1$_{LONG}$ (humanized codons and artificial Kozak sequences), or the polycistronic plasmid vL1 (encodes all three viral L1 ORFs and Kozak sequences). (**A**) Expression of L1 isoforms was visualized with mAb 2D11 (only recognizing conformational L1 epitopes), serum mix from five tumor-bearing animals and anti-HA as a control. (**B**) Western blotting reveals two high-MW bands only in L1$_{SHORT}$–expressing cells corresponding to L1 dimers (#) and trimers (§). Transfection with vL1 results in synthesis of both L1$_{LONG}$ as well as L1$_{SHORT}$ and its multimer. (**C**) Upon harsher denaturation of cell lysates, L1$_{SHORT}$ multimer bands disappear. Vinculin served as loading control.

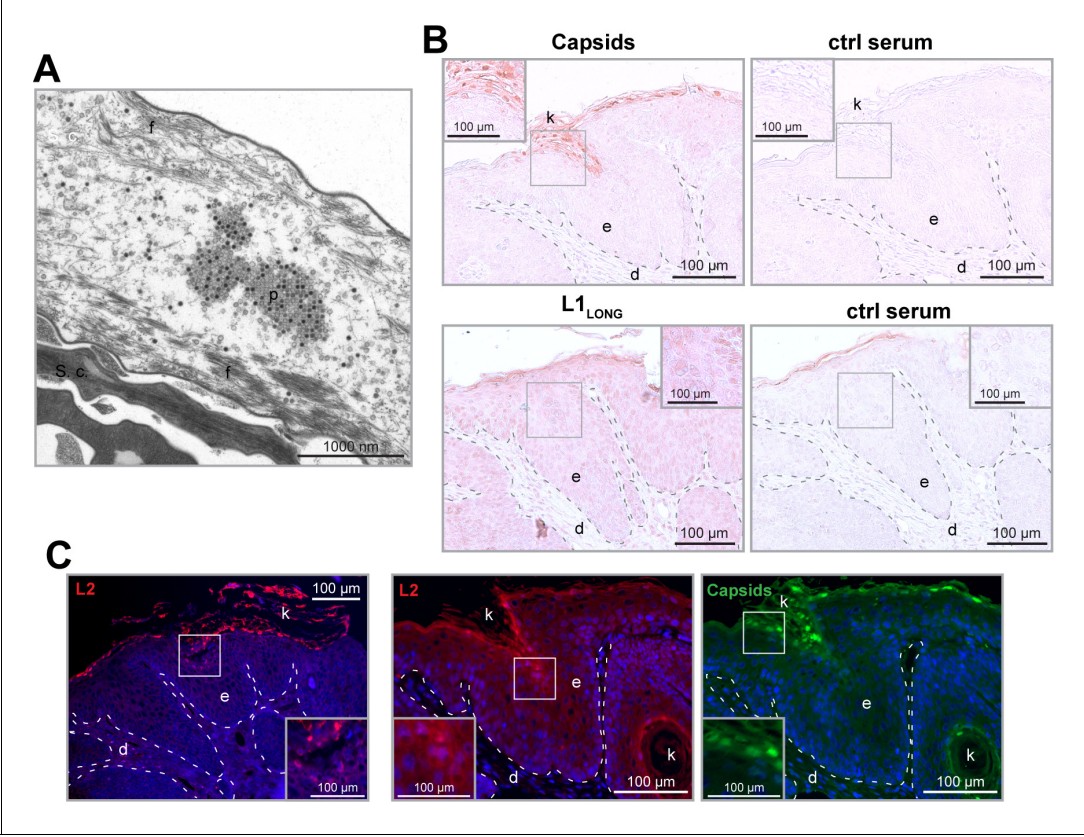

**Figure 10.** L1$_{LONG}$ synthesis occurs much earlier than capsid formation in vivo. (**A**) EM micrograph of MnPV particles (p) in the *stratum corneum* (s.c.) of a MnPV-induced papilloma. The nearly shed cell shown here is strongly degraded and only tonofilaments (f) are left. (**B**) Immunohistochemical analysis reveals MnPV capsids (detected with serum of a MnPV-VLP-immunized *Mastomys*) in the uppermost layers of the *stratum corneum* while L1$_{LONG}$ (detected by serum from a mouse immunized with the N-terminal 31 residues of MnPV-L1$_{LONG}$) appears throughout the whole epidermis. Pre-immune sera were used as controls. (**C**) Tissue sections were stained with cross-reactive anti-HPV-L2 antibody (K18L2, red) or cross-reactive anti-HPV-VLP guinea pig serum (green) as controls. Consistent with capsid formation, L2 only appears in the uppermost layers of the tissue shortly prior to VLP formation (d: dermis, e: epidermis, k: keratin).

*2018*), they represent a unique virus-host system to investigate the humoral response against MnPV L1 isoforms during the natural course of infection.

Of note was the finding that seroconversion against the isoforms L1$_{LONG}$ and L1$_{MIDDLE}$ appeared strikingly earlier than for L1$_{SHORT}$ (*Figure 2*), raising the question of the selective advantage for the virus and in turn its permissive cycle. To address this issue, pseudovirion-based infection assays were performed. Here, only sera binding L1$_{SHORT}$ inhibited pseudovirion infection, while sera recognizing the L1$_{LONG}$ and L1$_{MIDDLE}$ isoforms were not neutralizing (*Figure 4*). Furthermore, in contrast to L1$_{SHORT}$, the longer isoforms were also not able to form intact viral particles (*Figure 8*).

Conversely, the time course of MnPV E2 seroreactivity (*Figure 2D*) in comparison to the appearance of neutralizing antibodies (*Figure 4*) is indicative for viral replication and spread of infection (*Schäfer et al., 2011*; *Xue et al., 2010*). It is therefore reasonable to assume that the delay of neutralization capacity apparently allows the virus to accumulate. Pointing in this direction is also the kinetics of seroconversion against the L2 minor capsid protein (*Figure 2E*), finally assembling with L1 to form new infectious progeny virions at the end of the permissive cycle. Moreover, the correlation of the different ELISAs showed an obvious consistency between VLP and GST-L1$_{SHORT}$ that is not maintained when VLP or GST-L1$_{SHORT}$ are compared with GST-L1$_{LONG}$ (*Figure 3*).

As reported from mice immunized with recombinant HPV VLPs, neutralizing antibodies act via binding to linear or conformational epitopes (*Combita et al., 2002*; *Senger et al., 2009a*; *Zhang et al., 2016*). Although most high-affinity neutralizing antibodies bind to conformation-dependent epitopes, antibodies recognizing linear epitopes can be raised against improperly

maturated VLPs (used for immunization) or virions (in the course of a natural infection) (*Senger et al., 2009a*). Similar to HPV16 L1 (*Bissett et al., 2016*; *Zhang et al., 2016*), immunodominant epitopes could be identified within the DE and FG loops of MnPV L1 (*Figure 5*), known to form highly immunogenic conformational epitopes on the surface of virions (*Li et al., 2017*; *Zhang et al., 2016*).

Interestingly, we identified a novel epitope in the N-terminal region of L1, spanning at least some of the residues exclusive for $L1_{LONG}$ and extending to the first amino acids from $L1_{SHORT}$. Seroconversion against this peptide (referred to as $L1_{LONG}$aa1-41 in *Figure 6B*) followed roughly the same kinetics as $L1_{LONG}$ (*Figure 2B*) and most $L1_{LONG}$-positive sera recognized these 41 residues (*Figure 6D*). However, all sera only positive for $L1_{LONG}$, but negative for $L1_{SHORT}$ did not neutralize PsVs (*Figure 4C*) and further lost their ability to bind to $L1_{LONG}$ when the antigen was denatured prior to the ELISA (*Figure 7D*). Hence, the addition of N-terminal residues to $L1_{SHORT}$, obviously leads to a distinct folding of $L1_{LONG}$, thereby forming a new conformational epitope, which is not present in $L1_{SHORT}$-derived VLPs and natural virions that do not induce such antibodies upon immunization or infection (*Figure 6C*). Due to the fast kinetics of seroconversion against $L1_{LONG}$, this conformational epitope is apparently the predominant immunogenic structure in $L1_{LONG}$ and most probably masks neutralization epitopes that would be accessible in VLPs (*Figures 2F* and *4A*).

Indeed, in the case of HPV16 L1, the length of the N-terminus is decisive for efficient and correct assembly of L1 into VLPs (*Chen et al., 2000*). Consistently, neither MnPV $L1_{LONG}$ nor $L1_{MIDDLE}$ formed particles in the right size and shape in insect cells (*Figure 8*), a finding also reported for the mouse papillomavirus MmuPV L1 (*Joh et al., 2014*). Moreover, while co-expression of MnPV $L1_{SHORT}$ and L2 results in well-shaped infective pseudovirions, $L1_{LONG}$ and $L1_{MIDDLE}$ strikingly inhibited pseudovirion formation (*Figure 8*), suggesting that the N-terminus is literally hindering the assembly of both isoforms to capsids. Previously, crystallization of HPV L1 for 3D structure determination was not successful with the 'usual' L1 (being $L1_{SHORT}$) but required its N-terminal truncation (*Chen et al., 2000*), probably because the crystallization conditions lead to a folding of the N-terminus that prevents particle formation. Therefore, even the most complete 3D model 3J6R for HPV16 L1 (*Cardone et al., 2014*) lacks eight N-terminal residues. It was speculated that the N-terminus of L1 fills a gap in interpentameric structures (*Figure 8—figure supplement 2*) and since N-terminal deletion of eight residues has such a big impact on particle formation (*Chen et al., 2000*) it is likely that the additional 31 residues do not fit into this gap and furthermore completely sterically hinder L1 assembly. Additionally, the extended N-terminus not only inhibits assembly, but also causes a complete new folding of L1, thereby exposing a novel immunogenic conformational epitope that induces non-protective antibodies.

Nonetheless, using *Mastomys*-derived cells as genuine in vitro system to monitor expression of the three MnPV L1 variants (*Figure 9*), we found that $L1_{SHORT}$ but not the larger isoforms are recognized by mAb 2D11, confirming the presence of conformational epitopes found in capsomers. Importantly, this experiment also showed that $L1_{LONG}$ and $L1_{SHORT}$ ORFs are functional in a polycistronic setting and can be translated in *Mastomys* cells.

PV late gene expression is tightly regulated at transcriptional and post-transcriptional level (*Graham, 2017*). The MnPV transcription map obtained from skin lesions revealed L1 transcripts to be the most abundant in productive infections, which are encoded by polycistronic mRNAs (coding for E1E4, L2 and L1) exclusively controlled by the late viral promoter (*Salvermoser et al., 2016*). However, speculating about the mode of L1 isoform regulation, the following scenario could be envisioned: late PV transcripts are known to be strongly upregulated upon keratinocyte differentiation. However, due to insufficient suppressed late viral promoter activity, late transcripts can already be detected in undifferentiated cells (*Songock et al., 2017*). Indeed, in contrast to mucosal types (e.g. HPV11, 16 and COPV), cutaneous papillomaviruses (e.g. HPV1, 2 and BPV1) initiate their late functions in the lower epithelial regions (*Peh et al., 2002*). Consequently, this could lead to synthesis of the highly immunogenic $L1_{SHORT}$ already in early stages of infection, giving rise to neutralizing antibodies, thereby counteracting the permissive cycle and in turn the viral spread. However, the predicted presence of a stronger Kozak sequence in a upstream ORF from the conventional L1 ($L1_{SHORT}$) (*Supplementary file 3*), generates an alternative longer isoform of L1 which is expressed already in basal and suprabasal layers of MnPV-induced lesions (*Figure 10*) and similar to the pattern of L1 mRNA expression in MmuPV1-induced papillomas (*Xue et al., 2017*). This $L1_{LONG}$ isoform is also immunogenic but does not raise neutralizing antibodies.

Conversely and in agreement with other studies (*Biryukov and Meyers, 2015*; *Borgogna et al., 2014*), capsid formation could only be found in the granular and uppermost layers (*Figure 10*), which are less accessible for immune effector cells. However, based on in silico prediction, the capsid-forming L1$_{SHORT}$ cannot be efficiently translated from polycistronic transcripts coding for E1E4, L2 and L1 (*Supplementary file 3*), reinforcing the existence of a transcript in which the splicing acceptor is located immediately upstream of the L1$_{SHORT}$ AUG (*Salvermoser et al., 2016*). This splicing event is more tightly regulated than late promoter activity, based on a splicing silencer sequence suppressing the use of this site during mRNA processing (*Zhao et al., 2004*). Thus, only the combination of splicing regulation and the presence of favored ORFs ensure L1$_{SHORT}$ synthesis in upper layers of the epithelium (*Zhao et al., 2004*). This may allow a better escape from immune surveillance until the regular permissive cycle is completed.

Finally, the question remains why only certain PVs produce an immunogenic L1 isoform that is not needed for its own life cycle. Considering a virus-host interaction as consequence of an evolutionary process, there is a sophisticated balance between host immune surveillance and immune escape by the respective virus to ensure virus progeny production (*French and Holmes, 2020*). The temporal and spatial change of such an equilibrium in favor to infection determines the efficiency of viral accumulation and maturation as well as the spread to infect another host (*Rothenburg and Brennan, 2020*).

The occurrence of several L1 ORFs is not a peculiarity of MnPV, but also found in different HPV genera as well as in animal papillomaviruses such as MmuPV (*Joh et al., 2014*; *Webb et al., 2005*). It is intriguing that alternative ORFs are mostly present in mucosal 'high-risk' HPV types that cause clinical symptoms, but not in 'low-risk' types such as HPV6 and 11 (*Webb et al., 2005*). Hence, there must be a selection pressure to maintain several L1 initiation codons in these PVs types, thereby allowing the synthesis of different isoforms to get an advantage for the virus.

Naturally occurring mutations within neutralizing epitopes that reduce the antigenicity of HPV L1 and L2 proteins may contribute to humoral immune evasion (*Seitz et al., 2013*; *Yang et al., 2005*). Such L1 variants, isolated from premalignant HPV16 positive cervical tissue, showed impaired viral capsid assembly that can influence both B cell class switching and the production of non-neutralizing antibodies (*Yang et al., 2005*). Indeed, assembly-defective HPV16 VLPs impair the activation of dendritic cells that play a decisive role in activating adaptive immunity (*Yang et al., 2005*). Whether this also explains the late development of HPV16 L1 neutralizing antibodies (eight and nine months after the first positive HPV DNA detection) (*Gutierrez-Xicotencatl et al., 2016*) remains to be elucidated.

In conclusion, it is tempting to speculate that our results show that early synthesis of alternative immunogenic L1 isoforms represents a novel mode of humoral immune escape mechanism, favoring persistent infections and viral spread due to a delay of immune recognition by the host.

## Materials and methods

### Key resources table

| Reagent type (species) or resource | Designation | Source or reference | Identifiers | Additional information |
|---|---|---|---|---|
| Genetic reagent (*Mastomys coucha*) | African Multimammate rodent | DKFZ, Prof. F. Rösl | *Mastomys coucha* | Used as experimental model, **Hasche and Rösl, 2019** |
| Gene (*Mastomys natalensis Papillomavirus1*) | MnPV | GenBank | NC_001605.1 | **Tan et al., 1994** |
| Biological sample (*Mastomys coucha*) | Sera | This paper | | Sera tested for seroconversion against different MnPV proteins |

*Continued on next page*

*Continued*

| Reagent type (species) or resource | Designation | Source or reference | Identifiers | Additional information |
|---|---|---|---|---|
| Biological sample (*Mastomys coucha*) | Sera | *Vinzón et al., 2014* | | Sera from VLP-vaccinated animals |
| Cell line (*Mastomys coucha*) | MaFi132; *Mastomys coucha*- derived fibroblasts | DKFZ, Prof. F. Rösl | | *Hasche et al., 2016* |
| Cell line (*Homo sapiens*) | HeLaT | DKFZ, Prof. M. Müller, *Sehr et al., 2002* | HeLaT clone-4, | Used for pseudovirion–based neutralization assay |
| Cell line (*Homo sapiens*) | 293TT | DTP, DCTD TUMOR REPOSITORY | NCI-293TT; RRID:CVCL_1D85 | Used for pseudovirion production |
| Cell line (*Spodoptera frugiperda*) | Sf9 | DKFZ, Prof. M. Müller | RRID:CVCL_0549 | Insect cells, used for VLP production |
| Cell line (*Trichoplusia ni*) | TN-High Five | Gibco | BTI-TN-5B1-4; RRID:CVCL_C190 | Insect cells, used for VLP production |
| Strain, strain background (*Escherichia coli*) | TOP10 (DH10B) | Invitrogen | Cat#: C404010 | Chemically competent cells |
| Strain, strain background (*Escherichia coli*) | DH10MultiBac$^{Cre}$ | Geneva Biotech | Electrocompetent cells | *Fitzgerald et al., 2006* |
| Transfected construct (*Mastomys natalensis Papillomavirus1*) | pFBDM_L1$_{SHORT}$, pFBDM_L1$_{MIDDLE}$ , pFBDM_L1$_{LONG}$ | This paper | Backbone RRID:Addgene_110738 | Multibac constructs to transfect and express MnPV L1 variants in insect cells for VLP production |
| Transfected construct (*Mastomys natalensis Papillomavirus1*) | pPK-CMV-E3_L1$_{SHORT}$, pPK-CMV-E3_L1$_{MIDDLE}$, pPK-CMV-E3_L1$_{LONG}$ | This paper | | Constructs to transfect humanized ORFs and express MnPV L1 variants in MaFi132 cells |
| Transfected construct (*Mastomys natalensis Papillomavirus1*) | pPK-CMV-E3_vL1 | This paper | | Construct to transfect and express all L1 ORFs as found in the genuine MnPV genome in MaFi132 cells |
| Biological sample (*Mastomys natalensis Papillomavirus1*) | MnPV VLPs | This paper | | MnPV virus-like particles used for VLP-ELISA and assembly studies |
| Biological sample (*Mastomys natalensis Papillomavirus1*) | MnPV PsVs | This paper | | MnPV pseudovirions for infectivity assay and PBNA |

*Continued on next page*

*Continued*

| Reagent type (species) or resource | Designation | Source or reference | Identifiers | Additional information |
|---|---|---|---|---|
| Antibody | anti-L1 (Mouse monoclonal) | This paper | mAb 2E2, 2D11, 3H8, 2D6, 5E5 | ELISA (1:20-1:14,580), IF (1:5), WB (2D11, 1:1000) |
| Antibody | *Mastomys* serum mix (*Mastomys coucha* polyclonal serum) | This paper | | IF (1:1000), Peptide Array (1:300) |
| Antibody | Anti-HA clone 3F10 (Rat monoclonal) | Sigma-Aldrich | Cat#: 11867423001; RRID:AB_390918 | IF (1:1000), WB (1:1000) |
| Antibody | Anti-Vinculin clone 7F9 (Mouse monoclonal) | Santa Cruz | Cat#: sc-73614; RRID:AB_1131294 | WB (1:4000) |
| Antibody | Anti-L1$_{LONG}$aa1-31 serum (Mouse polyclonal serum) | This paper | | IHC (1:100) |
| Antibody | Anti-L2 serum (Guinea pig polyclonal serum) | DKFZ, Prof. M. Müller | | IHC (1:200) |
| Antibody | Anti-L2 clone K18L2 (Guinea pig polyclonal serum) | DKFZ, Prof. M. Müller, *Rubio et al., 2011* | | IHC (1:200) |
| Antibody | Anti-Mouse IgG (H+L), HRP Conjugate (Goat polyclonal) | Promega | Cat#: W4021; RRID:AB_430834 | ELISA (1:10,000), WB (1:10,000) |
| Antibody | Peroxidase AffiniPure Goat Anti-Rat IgG (H+L) | Jackson ImmunoResearch | Cat#: 112-035-003; RRID:AB_2338128 | WB (1:10,000) |
| Antibody | Goat anti-Mouse IgG (H+L), Alexa Fluor 488 (Goat polyclonal) | Invitrogen | Cat#: A11029; RRID:AB_138404 | IF (1:1000), IHC (1:1000) |
| Antibody | Goat anti-Guinea Pig IgG (H+L), Alexa Fluor 488 (Goat polyclonal) | Invitrogen | Cat#: A11073; RRID:AB_2534117 | IHC (1:1000) |
| Antibody | Donkey anti-Rat IgG (H+L), Alexa Fluor 488, (Donkey polyclonal) | Invitrogen | Cat#: A21208; RRID:AB_141709 | IF (1:1000) |
| Antibody | Mouse IgG (H and L) Antibody DyLight 680 Conjugated | Rockland Immunochemicals | Cat#: 610-144-121; RRID:AB_1057546 | Peptide Array (1:5000) |
| Recombinant DNA reagent | pPK-CMV-E3 | Promocell | Cat#: PK-MB-P003300 | |
| Peptide, recombinant protein | GST-L1$_{SHORT}$, GST-L1$_{MIDDLE}$, GST-L1$_{LONG}$, | This paper, *Schäfer et al., 2010* | | GST protein fused to the different MnPV L1 variants |

*Continued on next page*

*Continued*

| Reagent type (species) or resource | Designation | Source or reference | Identifiers | Additional information |
|---|---|---|---|---|
| Peptide, recombinant protein | GST-L1$_{LONG}$aa1-31, GST-L1$_{LONG}$aa1-41 | This paper | | GST protein fused to the N-terminus of MnPV L1 |
| Peptide, recombinant protein | GST-E2, GST-L2 | This paper, *Schäfer et al., 2010* | | GST protein fused to MnPV E2 or L2 |
| Peptide, recombinant protein | L1 peptide array | PEPper PRINT GmbH | | *Stadler et al., 2008* |
| Commercial assay or kit | Dako REAL Detection System, Peroxidase/AEC, Rabbit/Mouse | Agilent | Cat#: K5007 | IHC chromogenic detection Kit |
| Commercial assay or kit | Gaussia glow juice | PJK Biotech | Cat#: 102542 | Luciferase activity detection kit |
| Chemical compound, drug | DAPI | Sigma-Aldrich | Cat#: D9542-5MG | |
| Software, algorithm | GraphPad Prism 6.0 | GraphPad | | |

## Animals

The *Mastomys coucha* breeding colony naturally infected by MnPV is maintained under SFP conditions in individually ventilated cages (Tecniplast GR900) at 22+/- 2°C and 55+/- 10% relative humidity in a light/dark cycle of 14/10 hr. *Mastomys* were fed with mouse breeding diet and allowed access to water ad libitum. For the follow-up experiment, animals were monitored for the duration of their lifetime until they had to be sacrificed due to tumor development or decrepitude. Blood was taken in intervals from 2–8 weeks by puncturing the submandibular vein of anesthetized animals (3% isoflurane), starting at the age of eight weeks.

## Cell culture conditions

293TT, HeLaT and MaFi132 cells were grown in DMEM supplemented with 10% fetal calf serum (FCS), 1% Penicillin/Streptomycin and 1% L-glutamine. Media of HeLaT and 293TT cell were further supplemented with Hygromycin B (125 µg/ml) to maintain additional SV40 large T-antigen expression. All cell lines were kept at 37°C, 5% $CO_2$ and 95% humidity and regularly checked for Mycoplasma via PCR. Sf9 and TN-High Five insect cells were kept as described elsewhere (*Senger et al., 2009b*).

## GST-capture ELISA

As previously described (*Schäfer et al., 2011*; *Schäfer et al., 2010*), glutathione-casein was diluted in 50 mM carbonate buffer (pH9.6) and 200 ng/well were coated overnight at 4°C to 96 well plates (Nunc PolySorp). After blocking with 180 µl/well casein blocking buffer (CBB, 0.2% casein in PBST: 0.05% Tween-20 in PBS) for 1 hr at 37°C, the plate was incubated with the respective antigen (bacterial lysate containing the GST-antigen-SV40-tag fusion protein) for 1 hr at RT. *Mastomys* sera diluted 1:50 in CBB containing GST-SV40-tag were incubated for 1 hr at RT to remove unspecific reaction against bacterial proteins or the GST-SV40-tag fusion protein. Afterwards, ELISA plates were washed four times with PBST and incubated with pre-incubated sera for 1 hr at RT. After washing four times, 100 µl/well HPR-conjugated goat anti-mouse IgG (H+L) antibody (Promega, 1:10,000 in CBB) were applied for 1 hr at RT. Antibodies were quantified colorimetrically by incubating with 100 µl/well substrate buffer for 8 min (0.1 mg/ml tetramethylbenzidine and 0.006% $H_2O_2$ in 100 mM sodium acetate, pH6.0). The enzymatic reaction was stopped with 50 µl/well 1 M sulfuric acid. The

absorption was measured at 450 nm in a microplate reader (Labsystems Multiskan, Thermo Fisher Scientific). To calculate the serum reactivity against the respective antigen, sera were tested in parallel against the GST-SV40-tag fusion protein and the reactivity was subtracted from the reactivity against the GST-antigen-SV40-tag. Each ELISA was performed in duplicates at least. The cut-offs were calculated individually for each antigen by measuring sera of virus-free animals.

## VLP-ELISA

VLP-ELISAs were performed as described elsewhere (*Vinzón et al., 2014*). Briefly, 100 ng/well purified high quality L1$_{SHORT}$-VLPs were coated overnight in 50 mM carbonate buffer pH9.6 and blocked with CBB the next day. After incubation for 1 hr at RT with three-fold dilutions of sera in CBB, plates were washed four times with PBST and incubated with goat anti-mouse IgG-HRP (1:10,000 in CBB). After four washes, color development and measurement was performed as described for the GST-ELISA. Antibody titer represents the last reciprocal serum dilution above the blank.

## Denatured ELISAs

VLPs and GST fusion antigens were denatured at 95°C for 10 min in coating buffer (50 mM carbonate buffer pH9.6) prior to coating to ELISA plates overnight at 37°C. For the denatured VLP-ELISA, further steps were carried out according to the VLP-ELISA protocol described above. Denatured GST-ELISA antigens were then directly coated onto the ELISA plates overnight at 37°C and further steps (blocking, washing, incubation with sera, color reaction) were carried out according to the GST-ELISA protocol. The five monoclonal MnPV anti-L1$_{SHORT}$ antibodies mAb2E2, mAb2D6, mAb2D11, mAb5E5 and mAb3H8 (reactivities shown in *Supplementary file 2*) were generated via hybridoma technique from BALB/c mice vaccinated with MnPV L1$_{SHORT}$-VLPs and were used together with a *Mastomys* serum mix (sera from five tumor-bearing animals) as controls for denaturation conditions.

## Peptide arrays

The peptide array (PEPperPRINT GmbH, Germany) was produced as previously described (*Stadler et al., 2008*). The amino acid sequence of MnPV L1 was elongated with neutral GSGSGSG linkers at the C- and N-termini to avoid truncated peptides. Elongated antigen sequences were translated into 15 aa peptides with peptide-peptide overlaps of 14 aa. The resulting peptide microarray contained 530 different overlapping L1 peptides printed in duplicates and framed by additional HA (YPYDVPDYAG, 86 spots) control peptides. The peptide array was incubated for 10 min in PBST, followed by incubation in Rockland Blocking Buffer MB-070 (RBB; Rockland Immunochemicals, USA) for 1 hr. After short rinsing with PBST, the array was incubated for 16 hr at 4°C with *Mastomys* serum mix at a dilution of 1:300 in 10% RBB in PBST. The array was washed three times for 1 min with PBST and then incubated for 1 hr at RT with 0.2 µg/ml 10% RBB in PBST goat anti-mouse IgG (Fc) DyLight680 (Rockland Immunochemicals, USA). Subsequently, the array was washed three times for 1 min with PBST and rinsed with 1 mM TRIS-HCL pH7.4. As peptide controls, HA peptide spots were stained with monoclonal mouse-anti-HA IgG antibody (12CA5, kindly provided by Dr. G. Moldenhauer, DKFZ) conjugated with DyLight800 (Lightning-Link, Innova Biosciences, UK), followed by washing as described above. The antibody was diluted to 1 µg/ml in 10% RBB in PBST and staining was performed for 1 hr at RT in the dark followed by washing as described above. After drying of the array, fluorescence images were acquired with an Odyssey Infrared Imager (LICOR, USA) at a resolution of 21 µm. Scanner sensitivity was set to 7.0 for the 700 and 800 nm channels respectively, the focal plane was set to +0.8 mm. Quantification of spot intensities, based on 16-bit gray scale tiff files and microarray image analysis, via PepSlide Analyzer (SICASYS Software GmbH, Germany). A software algorithm breaks down fluorescence intensities of each spot into raw, foreground and background signal, and calculates averaged median foreground intensities and spot-to-spot deviations of spot duplicates. Averaged spot intensities of the assays with the sample were plotted against the antigen sequence from N- to C-terminus to visualize overall spot intensities and signal-to-noise ratios (intensity plot).

## Pseudovirion production

Pseudovirions were produced as previously described (*Buck and Thompson, 2007*). 293TT cells were co-transfected by calcium phosphate transfection with plasmids encoding humanized MnPV L1

isoforms (L1$_{LONG}$, L1$_{MIDDLE}$ and L1$_{SHORT}$), L2 and a reporter plasmid encoding Gaussia luciferase. The 2$^{nd}$ and 3$^{rd}$ ATG of L1$_{LONG}$ and the 2$^{nd}$ ATG of L1$_{MIDDLE}$ were mutated to GCG (alanine) to exclusively guarantee L1$_{LONG}$ or L1$_{MIDDLE}$ expression. Transfected cells were incubated for 48 hr, harvested and resuspended in an equal volume of PBS and supplemented with 0.5% Brij 58 (Sigma) and 1% RNase A/T1 mix (Thermo Fisher Scientific). Cells were lysed for 24 hr under rotation at 37°C to allow pseudovirion maturation prior to adjustment with 5 M NaCl to 0.85 M NaCl and treatment with 700 U Benzonase (Merck) for 1 hr at 37°C. For purification, the lysate was transferred on top of a three-step gradient of 27%, 33% and 39% Iodixanol (Optiprep, Sigma) diluted in 0.8 M NaCl/ DPBS. and centrifuged at 37,000 rpm for 5 hr at 16°C in a swinging bucket rotor. Fractions of 500 µl each were collected in siliconized LoBind tubes (Eppendorf) and quantity and quality of pseudovirions in each fraction was assessed by electron microscopy (EM) and Gaussia luciferase reporter activity after infection of HeLaT cells.

## Pseudovirion-based neutralization assay

As previously described (*Buck et al., 2005a*), animal sera (in duplicates, initial dilution 1:60 in medium) were subjected to 1:3 serial dilutions in 96-well cell culture plates (Greiner Bio-One GmbH). Then, the sera were mixed with 40 µl of diluted pseudovirions and incubated for 15 min at RT. Then, 50 µl of 2.5 × 10$^5$ HeLaT cells/ml were seeded to the pseudovirion-serum mixture and cultured for 48 hr at 37°C. The activity of secreted Gaussia luciferase was measured 15 min after adding coelenterazine substrate and Gaussia glow juice (PJK Biotech, Germany) according to the manufacturer's instructions in a microplate luminometer reader (Synergy 2, BioTek). The neutralization titer represents the reciprocal of the highest dilution that reduces the signal by at least 50%.

## Construction of pFBDM plasmids and Multibac plasmids

Two copies of MnPV wildtype L1$_{LONG}$, L1$_{MIDDLE}$ and L1$_{SHORT}$ were inserted into the Multibac vector pFBDM using *EcoRI/HindIII* and *XmaI/SphI*. Comparable to the VLP production, to ensure that only L1$_{LONG}$ and L1$_{MIDDLE}$ are expressed in the Multibac expression system, the 2$^{nd}$ and 3$^{rd}$ ATG start codons of L1$_{LONG}$ and the 2$^{nd}$ ATG of L1$_{MIDDLE}$ were mutated to GCG (alanine). The recombinant MultiBac bacmids were generated by electroporation (1.8 kV pulse) of DH10MultiBac$^{Cre}$ *E. coli* with the generated plasmids, followed by selection with antibiotics and blue/white screening (*Fitzgerald et al., 2006*). Recombinant MultiBac bacmids were isolated by QIAGEN Plasmid Mini Kit followed by ethanol precipitation.

## VLP production and purification

Recombinant baculoviruses were generated as previously described (*Vinzón et al., 2014*) with some modifications. One µg of MultiBac bacmid containing L1$_{LONG}$, L1$_{MIDDLE}$ or L1$_{SHORT}$ was diluted in 1 ml transfection buffer (25 mM Hepes, 125 mM CaCl$_2$, 140 mM NaCl, pH7.2) and added dropwise to Sf9 cells. After incubation at 27°C for 5 hr, cells were washed twice and then cultured for 6 days in supplemented TNM-FH medium (Sigma). One ml supernatant was used for generation of a high-titer baculovirus stock by infecting 2 × 10$^6$ Sf9 cells in a T25 flask followed by virus amplification for 6 days. This step was repeated with the obtained supernatant two times with increasing cell numbers (3 ml supernatant for 1 × 10$^7$ cells in a T75 flask and 5 ml supernatant for 2.5 × 10$^7$ cells in a T175 flask). TN-High Five cells were cultivated to a density of 2.5 × 10$^6$/ml in 250 ml suspension culture, which were then pelleted and resuspended in 42 ml EX-CELL 405 serum-free medium (Sigma) and 8 ml high titer virus stock. The cells were shaken at a low speed for 1 hr at RT and then incubated within 250 ml final volume of medium for 3 days at 27°C. Cell pellets were harvested by centrifugation (3000 rpm for 10 min at 4°C in a Sorvall GS-3 rotor) and washed in pre-chilled PBS for two times. Dry pellets were resuspended in 10 ml VLP extraction buffer (5 mM MgCl$_2$, 5 mM CaCl$_2$, 150 mM NaCl, 0.01% Triton X-100 and 20 mM Hepes pH7.4) containing 200 µl 100 mM PMSF, and then followed by three times sonication. A two-step gradient consisting of 7 ml of 40% sucrose on top of 7 ml CsCl solution was prepared. Clear cell lysate was obtained by centrifugation (10,000 rpm for 10 min at 4°C in a Sorvall F-28/50 rotor) and carefully loaded onto the top of the CsCl layer. After centrifugation (27,000 rpm for 3 hr at 10°C in SW-31Ti rotor), the interphase between sucrose and CsCl together with the complete CsCl layer was transferred into a Quickseal tube. The fractions were collected in 1 ml aliquot after 16 hr centrifugation at 48,000 rpm at 20°C in a Beckman 70Ti rotor and

analyzed by Coomassie blue dye and Western blot. Small aliquots from the fraction with highest and lowest protein yield were dialyzed against $H_2O$ on a membrane filter and analyzed by EM.

## Electron microscopy (EM)

VLP and PsV preparations or tissue were fixed with buffered aldehyde solution (2% formaldehyde, 2% glutaraldehyde, 1 mM $MgCl_2$, 2% sucrose in 100 mM calcium cacodylate, pH7.2), followed by post-fixation in buffered 1% $OsO_4$, graded dehydration with ethanol and resin-embedding in epoxide (12 g glycid ether, 6.5 g NMA, 6.5 g DDSA, 400 µl DMP30, all from Serva, Germany). Ultrathin sections at nominal thickness 60 nm and contrast-stained with lead-citrate and Uranylacetate were observed in a Zeiss EM 910 at 100 kV (Carl Zeiss, Oberkochen, Germany) and micrographs were taken with image-plates, scanned at 30 µm resolution (Ditabis micron, Pforzheim, Germany).

## Transfection of mammalian cells, SDS-PAGE and western blotting

Variants of L1 ORFs were cloned into pPK-CMV-E3 expression plasmids. For exclusive and strong eukaryotic expression of the respective L1 isoform, viral codons were humanized, unwanted start codons mutated (L1$_{SHORT}$ starts from 3$^{rd}$ ATG; L1$_{MIDDLE}$ starts from 2$^{nd}$ ATG, 3$^{rd}$ ATG mutated; L1$_{LONG}$ starts from 1$^{st}$ ATG, 2$^{nd}$ and 3$^{rd}$ ATG mutated) and the ORFs were cloned downstream of an artificial Kozak sequence. Alternatively, the complete genuine viral L1 ORF encoding start codons and Kozak sequences of all isoforms was cloned and termed pPK-CMV_MnPV-vL1.

MaFi132 cells (400,000 cells/10 cm dish) were transfected with 5 µg pPK-CMV_MnPV-L1$_{SHORT}$ or either 10 µg pPK-CMV_MnPV-L1$_{MIDDLE}$, pPK-CMV_MnPV-L1$_{LONG}$ or pPK-CMV_vL1 24 hr after seeding using TurboFect (Thermo Fisher Scientific) according to the manufacturer's protocol. Cells were collected 48 hr after transfection, washed in PBS and lysed for 30 min on ice in 1.25x Laemmli buffer (78 mM Tris pH6.8, 2.5% SDS, 6.25% glycerol, 0.125% bromophenol blue, 2.5% β-mercaptoethanol). Lysates were then heated at 95°C for 5 min, chilled on ice and treated with 100 U/ml Benzonase (Millipore) for 5 min at RT. Protein concentrations were measured using a NanoDrop spectrophotometer. Forty µg lysate/lane were loaded to 8% SDS-PAGE. To guarantee complete denaturation of the samples, additional DTT and β-mercaptoethanol were added to a final concentration of 100 mM and 4%, respectively, prior to incubation for 1 hr at RT and heating at 95°C for 10 min. After blotting, proteins were detected with anti-HA (3F10, 1:1000, Roche), anti-vinculin (7F9, 1:4000, Santa Cruz), anti-MnPV-L1$_{SHORT}$ (mAb 2D11, 1:5) or *Mastomys* serum mix (1:1000) prior to detection with goat anti-mouse-HRP (W4021, 1:10,000, Promega) or goat anti-rat-HRP (1:10,000, Jackson ImmunoResearch). For Coomassie staining, gels were incubated overnight in Coomassie stain and then destained in 20% methanol.

## Immunofluorescence stainings

MaFi132 cells were transfected with L1 isoforms as described above and seeded on glass cover slides after 24 hr. Additional 48 hr later the cells were washed with PBS and fixed for 10 min in 4% PFA. Cells were blocked in 10% goat serum/0.3% Triton X-100 in PBS for 1 hr and stained with anti-MnPV-L1$_{SHORT}$ (2D11, 1:5), *Mastomys* serum mix (1:1000) or anti-HA (3F10, 1:1000, Roche) and the respective secondary goat anti-mouse or donkey anti-rat IgG (conjugated to AlexaFluor488, 1:1000, Invitrogen). Nuclei were stained with DAPI. Cover slides were mounted with Faramount Aqueous Mounting Medium (Dako) and imaged with a Cell Observer (Carl Zeiss).

## Immunohistochemistry (IHC)

Staining of formalin-fixed, paraffin-embedded tumors was performed as previously described (*Hasche et al., 2017*). Briefly, deparaffinized sections were heated in citrate buffer pH6.0 prior to blocking with 5% goat serum/5% FCS/1% BSA in PBS and incubation with primary antibodies (serum of a VLP-vaccinated *Mastomys* (*Vinzón et al., 2014*), serum of a mouse immunized with the N-terminal 31 aa of MnPV-L1$_{LONG}$ in the OVX313 platform (*Spagnoli et al., 2017*) or the respective preimmune sera, cross-reactive anti-L2 (K18L2) (*Rubio et al., 2011*) or serum of a guinea pig immunized with Gardasil9 (unpublished) overnight at 4°C. Detection of L1 isoforms was achieved with the Dako REAL Detection System, Peroxidase/AEC, Rabbit/Mouse. The color reaction with AEC/$H_2O_2$ substrate solution (Sigma) was stopped with distilled water followed by counterstaining with hemalum solution (Carl Roth, Karlsruhe, Germany). Fluorescence stainings were detected with anti-mouse-

IgG1-Alexa594 or anti-guinea-pig-Alexa488 (Invitrogen) and nuclei were stained with DAPI. Sections were mounted with Dako Faramount Aqueous Mounting Medium.

## Statistical analysis

Data analyses and graphic representations were performed with GraphPad Prism 6.0 Software and the respective statistical test indicated in the figure legends at 95% confidence interval and an alpha level of 5% to assess significance. For time course analyses, individual time points were compared to the eight-week starting time point. The rate of $L1_{LONG}$- and $L1_{SHORT}$- positive animals was calculated and compared with a two-tailed McNemar's test at an alpha level of 5% to assess significance.

# Acknowledgements

We gratefully thank Dr. K Richter and Dr. M Neßling (Central Unit Electron Microscopy, DKFZ) for acquisition of EM images. We also highly appreciate helpful suggestions from Dr. T Holland-Letz (Division of Biostatistics, DKFZ). Support and assistance by the animal technicians and veterinarians of the Center for Preclinical Research, DKFZ is also highly acknowledged. This paper is dedicated to Prof. Vladimir Vonka (former Director of Experimental Virology, Charles University, Prague) on the occasion of his 90th birthday.

# Additional information

## Funding

| Funder | Grant reference number | Author |
| --- | --- | --- |
| China Scholarship Council | Graduate Student Fellowship | Yingying Fu
Rui Cao |
| Bundesministerium für Bildung und Forschung | 031L0095B | Rui Cao
Frank Rösl |
| Wilhelm Sander-Stiftung | 2018.093.1 | Miriam Schäfer
Daniel Hasche |
| Wilhelm Sander-Stiftung | 2010.019.1 | Sabrina E Vinzón
Frank Rösl |

The funders had no role in study design, data collection and interpretation, or the decision to submit the work for publication.

## Author contributions

Yingying Fu, Data curation, Formal analysis, Validation, Investigation, Visualization, Writing - original draft; Rui Cao, Data curation, Validation, Investigation, Visualization; Miriam Schäfer, Sonja Stephan, Ilona Braspenning-Wesch, Laura Schmitt, Investigation; Ralf Bischoff, Resources, Data curation, Formal analysis, Investigation, Visualization; Martin Müller, Resources; Kai Schäfer, Methodology; Sabrina E Vinzón, Conceptualization, Formal analysis, Investigation, Visualization, Methodology, Writing - original draft; Frank Rösl, Conceptualization, Resources, Supervision, Funding acquisition, Writing - original draft, Project administration; Daniel Hasche, Conceptualization, Data curation, Formal analysis, Supervision, Funding acquisition, Validation, Investigation, Visualization, Methodology, Writing - original draft, Project administration

## Author ORCIDs

Sabrina E Vinzón (iD) https://orcid.org/0000-0002-1330-9125
Daniel Hasche (iD) https://orcid.org/0000-0001-9306-3059

## Ethics

Animal experimentation: The animals are housed and handled in accordance with local (DKFZ), German and European statutes. All animal experiments were approved by the responsible Animal Ethics Committee for the use and care of live animals (Regional Council of Karlsruhe, Germany, File No 35-9185.81/G289/15).

## Decision letter and Author response

Decision letter https://doi.org/10.7554/eLife.57626.sa1
Author response https://doi.org/10.7554/eLife.57626.sa2

---

## Additional files

### Supplementary files

- Supplementary file 1. Alignment of L1 amino acid sequences of MnPV and HPV6, 16 and 18.
- Supplementary file 2. Binding properties of monoclonal antibodies against L1 isoforms.
- Supplementary file 3. Prediction of ATG usage in MnPV late transcript Q.
- Transparent reporting form

### Data availability

All data generated or analysed during this study are included in the manuscript and supporting files.

---

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
