## [Decision Letter]

**Acceptance summary:**

This study investigates the antibody response to the major coat protein L1 throughout the life cycle of a natural papillomavirus infection the multimammate mouse *Mastomys*. Three isoforms of L1 can be expressed L1 long, L1 middle, L1 short. This study and others confirm that these isoforms are differentially expressed both temporally and spatially; only L1 short is expressed late in the infectious cycle in the upper layers of the epithelium and elicits neutralising antibody. L1 long and middle are expressed early in the infectious cycle in the lower epithelial layers but induce predominantly non neutralising antibody. The authors propose that L1 long and middle antibody responses are a viral immune evasion strategy. While no mechanisms for this are described, the study is well-conceived, the findings are novel and unexpected and will provoke interest and further experimentation.

**Decision letter after peer review:**

Thank you for submitting your article "Expression of different L1 isoforms of *Mastomys natalensis* papillomavirus as mechanism to circumvent adaptive immunity" for consideration by *eLife*. Your article has been reviewed by three peer reviewers, including Margaret Stanley as the Reviewing Editor and Reviewer #3, and the evaluation has been overseen by Satyajit Rath as the Senior Editor. The following individual involved in review of your submission has agreed to reveal their identity: Neil Christensen (Reviewer #2).

The reviewers have discussed the reviews with one another and the Reviewing Editor has drafted this decision to help you prepare a revised submission.

Summary:

This study investigates the serological response to the major coat protein L1 throughout the life cycle of a natural papillomavirus infection the multimammate mouse *Mastomys*. Three alternative translation initiation codons of the L1 ORF are found in *Mastomys*. Depending upon which ATG is used three isoforms of L1 can be expressed L1 long, L1 middle, L1 short. This study and others confirm that these isoforms are differentially expressed both temporally and spatially; only L1 short is expressed late in the infectious cycle in the upper layers of the epithelium and elicits neutralising antibody. L1 long and middle are expressed early in the infectious cycle in the lower epithelial layers but induce predominantly non neutralising antibody. The authors propose that L1 long and middle antibody responses are a viral immune evasion strategy but no mechanisms for this are described. However the study is well-conceived, the findings are novel and unexpected and will provoke interest and further experimentation.

Revisions:

The reviews are favourable and the paper can be published in *eLife* but the authors should address the following comments and clearly identify in the text that their assertions that the responses they observe are an immune evasion mechanism is a speculation not an established fact.

1) In the correlation analyses in Figure 3 and 4, please indicate the animal's age for the sera used.

2) In Figure 4B, there appear to be only 8 data points, while the figure legend states that the sera from 60 animals were evaluated. Please reconcile.

3) Subsection “Mapping of immunodominant epitopes in MnPV L1”: Why were sera from tumor bearing animals used here? Is there any data on whether tumor-bearing and tumor free animals differ in virus antibody responses?

4) Figure 6 legend, last sentence: change to "GST-L1LONGaa1-41”, as indicate correctly in the figure.

5) Discussion paragraph seven: the N-terminus in long L1 must do more that block capsomer assembly because capsomers can induce neutralizing antibodies but long L1 cannot. It must induce a remarkable change in L1 monomer conformation.

6) Discussion paragraph nine: it would be helpful to clarify whether "cutaneous papillomaviruses” refers to just MnPV/MmuPV or also skin HPVs.

7) What are the authors views as to why a long/middle L1 is not capable of triggering a neutralizing ab response?

8) Figure 10B is not of publishable quality

---

## [Author Response]

Revisions:The reviews are favourable and the paper can be published in eLife but the authors should address the following comments and clearly identify in the text that their assertions that the responses they observe are an immune evasion mechanism is a speculation not an established fact.

Following this advice, we changed our conclusions accordingly:

“In conclusion, it is tempting to speculate that our results show that early synthesis of alternative immunogenic L1 isoforms represents a novel mode of humoral immune escape mechanism, favoring persistent infections and viral spread due to a delay of immune recognition by the host.”

1) In the correlation analyses in Figure 3 and 4, please indicate the animal's age for the sera used.

As already stated in Figure 3, all 682 sera were used and therefore represent the complete range of the animal age. However, we added the information:

“All graphs include the analysis of all 682 sera taken during the study”.

The correlation analyses in Figure 4 contain fewer animals that nevertheless again represent the complete age range. We added this information:

“Correlation analyses contain sera from animals representing the complete age range”.

2) In Figure 4B, there appear to be only 8 data points, while the figure legend states that the sera from 60 animals were evaluated. Please reconcile.

We understand the confusion which is attributable to a technical issue: The 60 sera were applied in serial dilution steps of one to three in both Pseudovirion-based Neutralization Assay (PBNA) and VLP-ELISA. Since the titers represent the last reciprocal serum dilution above the blank (for VLP-ELISA) and the reciprocal of the highest dilution that reduces the signal by at least 50% (for neutralization assay), several sera end up at the same titer. In the graph showing the correlation between the VLP-ELISA and the neutralization assay (Figure 4B), these 60 data points subsequently overlay, ending up with only eight visible points. However, to avoid confusion, we added following statement to the legend of Figure 4:

“Please note, that for both assays all 294 sera were diluted in three-fold dilution steps. Since the titers were calculated from the dilution, data points of different sera overlay when having the same titer in both assays.”

Furthermore, we also added the following sentence in the Materials and methods section.

“The neutralization titer represents the reciprocal of the highest dilution that reduces the signal by at least 50%.”

3) Subsection “Mapping of immunodominant epitopes in MnPV L1”: Why were sera from tumor bearing animals used here? Is there any data on whether tumor-bearing and tumor free animals differ in virus antibody responses?

The answer of the latter question is “yes” and this was the reason why sera from tumor-bearing animals were used. As shown in our previous studies (Schäfer et al., 2011, Vinzón et al., 2014), there is a quite obvious relationship of the higher levels of antibodies against L1_LONG/SHORT_ and the presence of skin lesions in which infectious progeny viruses are formed.

4) Figure 6 legend, last sentence: change to "GST-L1LONGaa1-41”, as indicate correctly in the figure.

We apologize for the typo and changed it to GST-L1_LONG_aa1-41 accordingly.

5) Discussion paragraph seven: the N-terminus in long L1 must do more that block capsomer assembly because capsomers can induce neutralizing antibodies but long L1 cannot. It must induce a remarkable change in L1 monomer conformation.

Yes, we mentioned that point already: “Hence, the addition of N-terminal residues to L1_SHORT_ obviously leads to a distinct folding of L1_LONG_, thereby forming a new conformational epitope which is not present in L1_SHORT_-derived VLPs and natural virions that do not induce such antibodies upon immunization or infection (Figure 6C).”

To emphasize this finding, we included the following statement: “Additionally, the extended N-terminus not only inhibits assembly, but also causes a complete new folding of L1, thereby exposing a novel immunogenic conformational epitope that induces non-protective antibodies”.

6) Discussion paragraph nine: it would be helpful to clarify whether "cutaneous papillomaviruses” refers to just MnPV/MmuPV or also skin HPVs.

We added this information:

“Indeed, in contrast to mucosal types (e.g. HPV11, 16 and COPV), cutaneous papillomaviruses (e.g. HPV1, 2 and BPV1) initiate their late functions in the lower epithelial regions (Peh et al., 2002)”.

7) What are the authors views as to why a long/middle L1 is not capable of triggering a neutralizing ab response?

As already stated (see point 5), our data suggest that an extended N-terminus abrogates L1 self-assembly and prevents the formation of repetitive, highly immunogenic structures usually found on native capsids. Consequently, an extended N-terminus of L1 leads to an alternative dominant immunogenic conformation, which is not present in infectious particles. Therefore, antibodies raised against this structure have no neutralization capability when PsVs as equivalents for mature MnPV virions were tested.

8) Figure 10B is not of publishable quality

Of course, this is due to embedding the figure in a.pdf-file. For publication, an original.tiff-file will be provided.